# An integrated multi-omics approach identifies the landscape of interferon-α-mediated responses of human pancreatic beta cells

Maikel L. Colli [1✉], Mireia Ramos-Rodríguez[2,3], Ernesto S. Nakayasu [4], Maria I. Alvelos[1], Miguel Lopes[1], Jessica L. E. Hill[5], Jean-Valery Turatsinze [1], Alexandra Coomans de Brachène[1], Mark A. Russell[5], Helena Raurell-Vila[2,3], Angela Castela[1], Jonàs Juan-Mateu[1], Bobbie-Jo M. Webb-Robertson[4], Lars Krogvold[6], Knut Dahl-Jorgensen[6], Lorella Marselli [7], Piero Marchetti[7], Sarah J. Richardson [5], Noel G. Morgan [5], Thomas O. Metz [4], Lorenzo Pasquali [2,3,8] & Décio L. Eizirik [1,9,10]

Interferon-α (IFNα), a type I interferon, is expressed in the islets of type 1 diabetic individuals, and its expression and signaling are regulated by T1D genetic risk variants and viral infections associated with T1D. We presently characterize human beta cell responses to IFNα by combining ATAC-seq, RNA-seq and proteomics assays. The initial response to IFNα is characterized by chromatin remodeling, followed by changes in transcriptional and translational regulation. IFNα induces changes in alternative splicing (AS) and first exon usage, increasing the diversity of transcripts expressed by the beta cells. This, combined with changes observed on protein modification/degradation, ER stress and MHC class I, may expand antigens presented by beta cells to the immune system. Beta cells also up-regulate the checkpoint proteins PDL1 and HLA-E that may exert a protective role against the autoimmune assault. Data mining of the present multi-omics analysis identifies two compound classes that antagonize IFNα effects on human beta cells.

[1] ULB Center for Diabetes Research, Medical Faculty, Université Libre de Bruxelles, Brussels 1070, Belgium. [2] Endocrine Regulatory Genomics, Department of Experimental & Health Sciences, University Pompeu Fabra, 08003 Barcelona, Spain. [3] Endocrine Regulatory Genomics Laboratory, Germans Trias i Pujol University Hospital and Research Institute, Badalona, Spain. [4] Biological Sciences Division, Pacific Northwest National Laboratory, Richland, WA 99352, USA. [5] Institute of Biomedical & Clinical Science, University of Exeter Medical School, Exeter EX2 5DW, UK. [6] Division of Pediatric and Adolescent Medicine, Faculty of Medicine, Oslo University Hospital, Oslo, Norway. [7] Department of Clinical and Experimental Medicine, Islet Cell Laboratory, University of Pisa, 56126 Pisa, Italy. [8] Josep Carreras Leukaemia Research Institute (IJC), Badalona, Barcelona, Catalonia, Spain. [9] WELBIO, Université Libre de Bruxelles, Brussels, Belgium. [10] Indiana Biosciences Research Institute, Indianapolis, IN, USA. ✉email: mcolli@ulb.ac.be

Type 1 diabetes (T1D) is a chronic autoimmune disease leading to pancreatic islet inflammation (insulitis) and progressive beta cell loss[1]. Type I interferons (IFN-I), a class of cytokines involved in antiviral immune responses[2], are involved in insulitis. Viral infections are a risk factor associated with T1D development[3] and individuals at risk of T1D show a type I interferon signature[4]. The type I interferon, interferon-α (IFNα), is expressed in islets of T1D patients[5], and antibodies neutralizing different isoforms of IFNα prevent T1D development in individuals with polyglandular autoimmune syndrome type 1[6]. Exposure of human pancreatic beta cells to IFNα recapitulates three key findings observed in human insulitis, namely HLA class I overexpression, endoplasmic reticulum (ER) stress and beta cell apoptosis[7].

Combination of genome-wide association studies (GWAS)[8] and studies using the ImmunoChip[9] have identified around 60 loci associated with the risk of developing T1D. Transcriptomic studies revealed that >70% of the T1D risk genes are expressed in human pancreatic beta cells[10], and many of these genes regulate innate immunity and type I IFN signaling[11].

Type I IFN signaling is often cell-specific, an effect mediated by differences in cell surface receptor expression, and activation of downstream kinases and transcription factors[12]. Thus, and considering the potential relevance of this cytokine to the pathogenesis of T1D, it is crucial to characterize its effects on human beta cells. To define the global impact of IFNα on human beta cells, we presently performed an integrative multi-omics analysis (ATAC-seq, RNA-seq and proteomics) of IFNα-treated human beta cells to determine the early, intermediate and late responses to the cytokine. The findings obtained indicate that IFNα promotes early changes in chromatin accessibility, activating distant regulatory elements (RE) that control gene expression and protein abundance. IFNα activates key transcription factors (TFs), including IRF1, which act as a mediator of the crosstalk between beta cells and immune cells via the expression of the checkpoint proteins PDL1 and HLA-E. Furthermore, IFNα induces modules of co-expressed mRNA and proteins that physically interact and have relevance to T1D pathogenesis. The integration of high-coverage RNA-seq and ATAC-seq indicates regulatory gene networks and reveals that alternative splicing and different first exon usage are key mechanisms expanding the repertoire of mRNAs and proteins expressed by stressed beta cells. Finally, mining the modules of co-expressed genes and the IFNα beta cell signature against the most recent catalogs of experimental and clinical drugs identifies two potentially interesting therapeutic targets for future trials.

## Results

**IFNα modifies beta cell mRNA expression similarly to T1D.** We performed a time course multi-omics experiment combining ATAC-seq, RNA-seq and proteomics of the human beta cell line EndoC-βH1 exposed or not to IFNα. The data were integrated to determine the dynamics of chromatin accessibility, gene/transcript expression and protein translation, respectively (Fig. 1a). We also performed RNA-seq of 6 independent human pancreatic islet preparations exposed or not to the cytokine at similar time points (Supplementary Fig. 1a). To assess whether our in vitro model is relevant for the in vivo islet inflammation (insulitis) in T1D, we took two approaches: (1) Examine whether candidate genes for T1D expressed in human islets are involved in IFN signaling (Supplementary Fig. 2a); and (2) Compare our in vitro data of IFNα-treated EndoC-βH1 cells and human islets with available RNA-seq data of human beta cells from T1D patients. In line with previous findings suggesting a role for IFNs on the pathogenesis of T1D[13], we found that T1D risk genes expressed

in human islets[10,14] are significantly enriched in immune-related pathways, including type I and II interferon regulation/signaling (Supplementary Fig. 2b). Next, we performed a Rank–Rank Hypergeometric Overlap (RRHO) analysis (which estimates the similarities between two ranked lists[15]) comparing the log₂ fold-change (FC) ranked list from RNA-seqs of EndoC-βH1 cells and human islets (IFNα-treated vs untreated) against an equally ranked list of genes obtained from RNA-seq of purified primary beta cells[16] from T1D and healthy individuals (Supplementary Fig. 2c and Supplementary Data 1). There was a significant intersection between upregulated genes induced by IFNα in both, EndoC-βH1 cells (362 overlapping genes) and human islets (850 overlapping genes), and genes induced by the local pro-inflammatory environment affecting primary beta cells from T1D individuals (Supplementary Fig. 2d, f). We also compared these two IFNα-treated datasets against beta cells from T2D patients[17], a condition mostly characterized by metabolic stress[18]. By contrast with the observations made in beta cells from T1D individuals, there was no statistically significant correlation between IFNα-regulated genes in EndoC-βH1 cells and human islets and the gene expression profile present in T2D beta cells (Supplementary Fig. 2e, g).

**IFNα induces early changes in chromatin accessibility.** The ATAC-seq experiments demonstrated that INFα induces early changes in chromatin accessibility, with >4400 regions of gained open chromatin regions (OCRs) detected at 2 h, which decreased to 1000 regions by 24 h (Fig. 1b and Supplementary Data 2); only nine regions had loss of chromatin accessibility (Fig. 1b). Most of the OCRs at 24 h were already modified at 2 h (fast response), and only 10% of OCRs were specifically gained at 24 h (late response). The gained OCRs were mostly localized distally to gene transcription starting sites (TSS) (Supplementary Fig. 3a) acting, therefore, as potential regulatory elements. These regions are evolutionary conserved (Supplementary Fig. 3b), and enriched for transcription factors (TFs) binding motifs (Supplementary Fig. 3c), including islet-specific TFs binding sequences.

To assess whether changes in chromatin remodeling were associated with variations in gene expression, we first quantified the frequency of ATAC-seq regions gained or stable in the proximity (40 kb window centered on the TSS) of genes with differential mRNA expression (up/down/non-regulated or non-expressed) (Supplementary Data 2). There was a higher proportion of upregulated genes associated with gained OCRs in comparison to stable regions at each time point analyzed (Fig. 1c). Moreover, the number of gained OCRs was associated with changes in both the proportion (Fig. 1d) and the intensity (Supplementary Fig. 3e) of transcript induction (Supplementary Fig. 3d, see Methods for more information). There was also a minor association between the number of stable regions and upregulated mRNAs at 2 h (Supplementary Fig. 3e), likely due to the activation of already nucleosome-depleted regions ahead of cytokine exposure[19]. Consistently with these results, there was an increase in the frequency of upregulated proteins coded by genes proximal to gained OCRs (Fig. 1e). Likewise, there was a progressive increase in IFNα-induced protein abundance depending on the number of linked gained open chromatin regions (Fig. 1f).

There was a strong correlation between upregulated mRNAs and induced proteins ($r^2$: 0.66 and 0.65 at, respectively, 8 and 24 h, $p < 2.2 \times 10^{-16}$) (Fig. 1g, first column), but a much lower similarity between downregulated mRNAs and proteins (Fig. 1g, second column). Gene ontology analysis of differentially abundant proteins upon IFNα treatment identified several biological processes involved in the pathogenesis of T1D, such

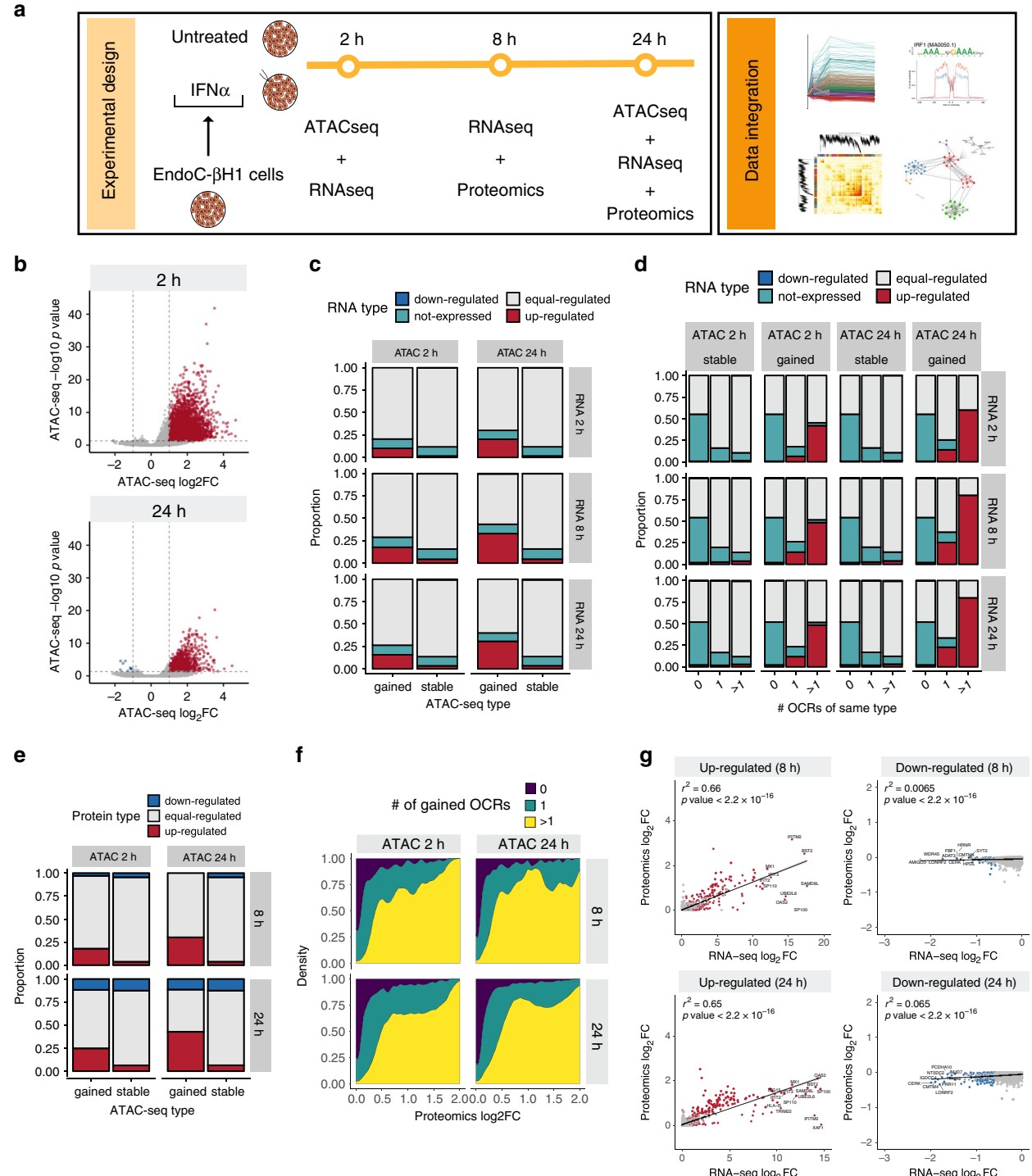

**Fig. 1 Exposure of EndoC-βH1 cells to interferon-α promotes changes in chromatin accessibility, which are correlated with gene transcription and translation. a** EndoC-βH1 cells were exposed or not to IFNα (2000 U/ml) for the indicated time points and different high-throughput techniques were performed to study chromatin accessibility (ATAC-seq, $n = 4$), transcription (RNA-seq, $n = 5$) and translation (Proteomics, $n = 4$). **b** Volcano plot showing changes in chromatin accessibility measured by ATAC-seq. Open chromatin regions indicated as gained (red) or lost (blue) had an absolute $\log_2$ fold-change ($|\log_2 FC|$) > 1, and a false discovery rate (FDR) < 0.05. The regions that did not reach such threshold were considered "stable" (gray). **c**, **d** Frequency of upregulated, downregulated or stable transcripts in the vicinity (<20 kb transcription start site (TSS) distance) of one or multiple open chromatin regions (OCRs) as classified in **b**. **e** Frequency of differentially abundant proteins in the vicinity (<20 kb TSS distance) of gained or stable open chromatin regions. **f** Distribution of IFNα-induced changes in protein abundance among upregulated proteins based on the number of linked gained OCRs. **g** Correlation between RNA-seq and proteomics of EndoC-βH1 cells exposed to INFα. The x axis represents the mRNA $\log_2 FC$. The most upregulated ($\log_2 FC > 0.58$, FDR < 0.05) and downregulated ($\log_2 FC < -0.58$, FDR < 0.05) mRNAs are filled in red and blue, respectively. The y axis indicates the proteomics $\log_2 FC$. The proteins most upregulated ($\log_2 FC > 0.58$, FDR < 0.15) or downregulated ($\log_2 FC < -0.58$, FDR < 0.15) are represented by red and blue borders, respectively. mRNAs and proteins not meeting these criteria were considered equal-regulated (gray fill and border, respectively).

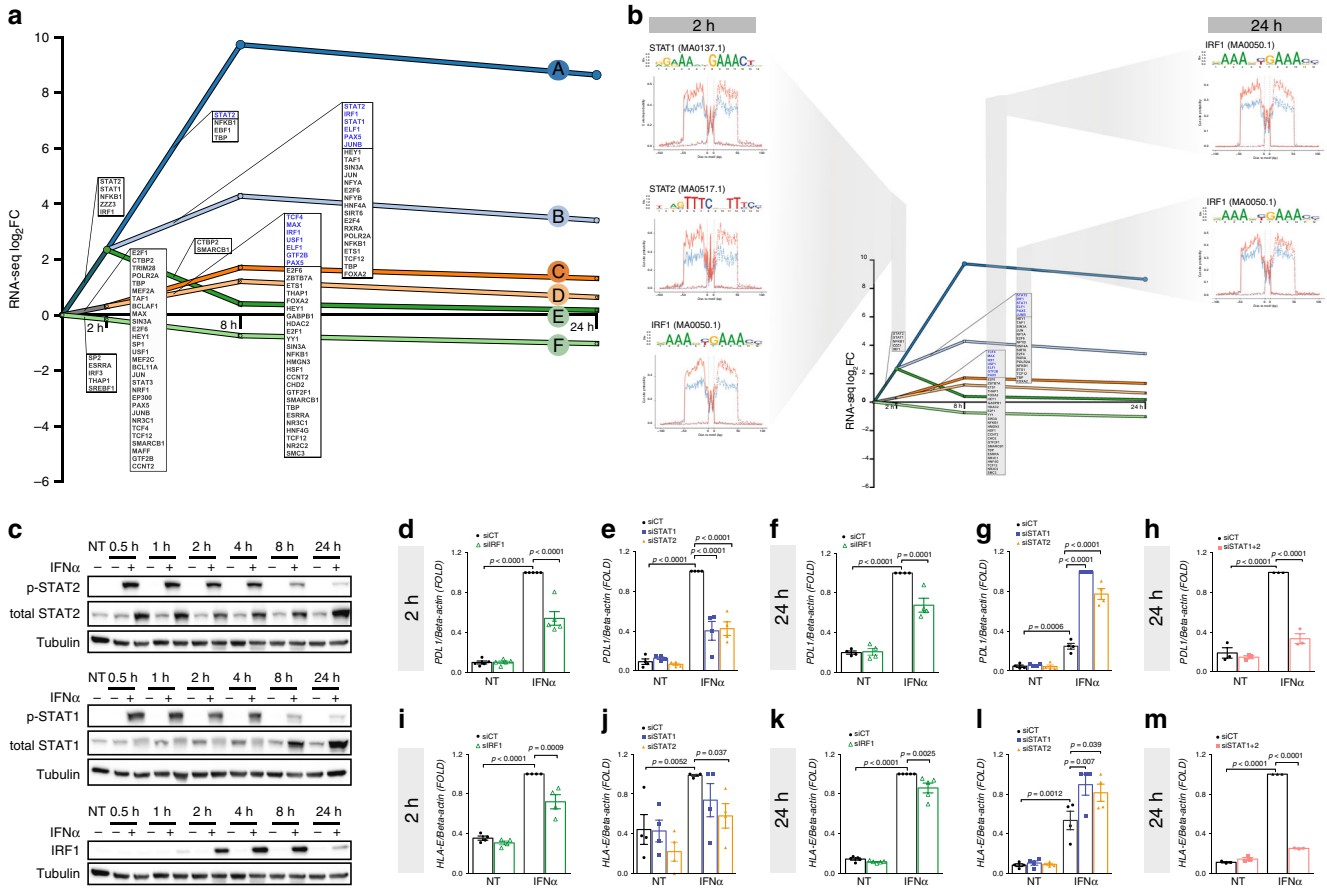

**Fig. 2 IRF1, STAT1 and STAT2 regulate IFNα-induced transcription and the expression of checkpoint proteins. a** The regulatory paths summarize the temporal patterns of the differentially expressed genes (DEG) detected by RNA-seq ($|log_2FC| > 0.58$ and FDR < 0.05, $n = 5$) (evaluated by DREM[25]). The $x$ axis represents the time and the $y$ axis the mRNA $log_2FC$. Each path corresponds to a set of co-expressed genes. Split nodes (circles) represent a temporal event where co-expressed genes diverge in expression. In blue are the TFs upregulated at the respective time points of the RNA-seq that may regulate the pathways. **b** IFNα promoted TFs footprint deepening in open chromatin regions (OCR) associated to genes from the indicated DREM pathways. OCRs were associated to the nearest gene TSS with a maximum distance of 1 Mb. Previously annotated TF matrices[79] were used to identify differential DNA-footprints induced by IFNα (blue lines = untreated cells, red lines = IFNα (24 h), dashed lines = reverse strand, continuous line = forward strand, Methods, $n = 4$). **c** Time course profile of STAT1, STAT2 and IRF1 protein activation in EndoC-βH1 cells exposed to IFNα (representative of four independent experiments). **d–m** EndoC-βH1 cells were transfected with an inactive control siRNA (siCT) or previously validated[7,24] siRNAs targeting IRF1 (siIRF1), STAT1 (siSTAT1), STAT2 (siSTAT2) or STAT1 plus STAT2 (siSTAT1 + 2). After 48 h the cells were exposed to IFNα The values were normalized by the housekeeping gene β-actin (mRNA) and then by the highest value of each experiment considered as 1 (for **h** and **m** ($n = 3$); for **e–g, i, j** and **l** ($n = 4$); for **d, k** ($n = 5$)), ANOVA with Bonferroni correction for multiple comparisons (**d–m**). Values are mean ± SEM (**d–m**). Source data are provided as Source Data file.

as antigen processing and presentation, responses to viruses, apoptosis and NK/T-cell responses (Supplementary Fig. 4a, b); groups of genes associated to protein modification and degradation were also present (Supplementary Fig. 4a, c). Furthermore, genes related to endoplasmic reticulum (ER) stress, another post-transcriptional mechanism that downregulates translation of many mRNAs[20], were also upregulated by IFNα at both the mRNA and protein levels (Supplementary Fig. 4d). These findings are in line with our previous observations[7] and were confirmed here in independent samples for two key ER stress markers, namely the transcription factor *ATF3*[21] and the ER chaperon *HSPA5* (also known as BiP/GRP78)[22] (Supplementary Fig. 4e–h). ER stress often decreases translation, which may explain the weak association observed between mRNA and protein expression in downregulated mRNAs and proteins (Fig. 1g).

**IRF1, STAT1 and STAT2 are key regulators of IFNα signaling**. To identify the key transcription factors involved, the expression

of differentially expressed genes (DEG) from all RNA-seq time points (Supplementary Data 3) was analyzed using the dynamic regulatory events miner (DREM) model[23]. This approach identified six patterns of co-expressed genes (Fig. 2a); 5 out of 6 pathways had an early peak of induction (2 or 8 h), which then decreased or remained stable until 24 h (Fig. 2a). The model compared the frequency of TF binding sites in the gene promoters between divergent branches of co-expressed genes, assuming that these TFs are responsible for the observed differences in gene expression profiles (Fig. 2a). This was compared with the TF occupancy determined by assaying the protection of the bound sequence to ATAC-seq transposase cleavage (footprint) (Supplementary Fig. 5a and Methods). There were footprints for the transcription factors IRF1, STAT1 and STAT2, which were deepened upon IFNα exposure in pathway B (which had the highest transcriptional upregulation at 2 h) and for IRF1 in two independent pathways, namely B and D at 24 h (Fig. 2b). Western blot analysis confirmed the activation of these TFs (Fig. 2c). STAT1 and STAT2 phosphorylation peaked between 0.5 and 1 h and then returned to near-basal levels at 24 h, while IRF1

peaked later, at 4–8 h decreased by 24 h (Fig. 2c); these findings support the observed TF footprint profiles (Fig. 2b). There was also a close correlation between DEGs induced by IFNα in RNA-seq of EndoC-βH1 cells and in human pancreatic islets (Supplementary Fig. 1b; $p < 2.2 \times 10^{-22}$ at 2, 8 and 24 h), which resulted in a similar pattern of gene activation under the control of analogous TFs (Supplementary Fig. 1c and Supplementary Data 4).

Individual DREM pathways usually regulate specific biological processes (GO) (Supplementary Fig. 5b, 1d). Among them, was the term "Regulation of immune responses" (Supplementary Fig. 5b). This pathway comprises several genes involved in the crosstalk between beta cells and the immune system, such as PDL1 (CD274), an immune checkpoint protein expressed in the islets of T1D individuals[24], and a second co-inhibitory molecule, HLA-E, recently identified as potential target for cancer immunotherapy[25] (Fig. 2d–m).

By using a previously validated siRNA targeting IRF1[24], we obtained around 60% knockdown (KD) of INFα-induced IRF1 protein and mRNA expression at 2 and 24 h (Supplementary Fig. 5c–f). IRF1 silencing led to a significant decrease in IFNα-induced PDL1 and HLA-E mRNA expression (Fig. 2d, f, i, k). Silencing of IRF1 also decreased IFNα-induced upregulation of the chemokines CXCL1 and CXCL10, the HLA-I component beta-2-microglobulin (B2M) and the suppressor of cytokine signaling 3 (SOCS3) (Supplementary Fig. 5d, f). Small interference RNAs targeting STAT1 (siSTAT1) or STAT2 (siSTAT2) promoted >70% KD of their respective proteins and mRNAs, (Supplementary Fig. 5g–j). Inhibiting STAT1 or STAT2 alone partially blocked the induction of PDL1 and HLA-E at 2 h (Fig. 2e, j), but led to a paradoxical increase in PDL1 and HLA-E expression at 24 h (Fig. 2g, i), which is probably due to a compensatory increase in expression of the non-targeted STAT[24]. In line with this, double KD of STAT1 + STAT2 led to downregulation of both PDL1 and HLA-E (Fig. 2h, m). STAT2 inhibition decreased the 2 h expression of IFNα-induced CXCL1/10, SOCS1 and MX1, whereas STAT1 KD only prevented CXCL10 induction (Supplementary Fig. 5h). At 24 h only 2 out of 4 genes remained partially inhibited by siSTAT2 (Supplementary Fig. 5j), whereas double KD of STAT1 + STAT2 prevented IFNα-induced gene upregulation at 24 h in most cases (Supplementary Fig. 5k).

Exposure of FACS-purified human beta cells (Supplementary Fig. 6a–c) to IFNα confirmed the upregulation of genes related to antigen presentation (HLA-I), antiviral responses (MX1, MDA5), ER stress (CHOP), immune cells recruitment (CXCL10) and checkpoint regulators (PDL1) (Supplementary Fig. 6d).

The checkpoint protein PDL1 is overexpressed in beta cells from people with T1D[24], and we presently evaluated the expression of another checkpoint protein, i.e. HLA-E[25]. IFNα upregulated HLA-E mRNA expression in EndoC-βH1 cells (Fig. 3a), dispersed human islets (Fig. 3b) and FACS-purified human beta cells (Fig. 3c) and augmented HLA-E protein expression in both EndoC-βH1 cells (Fig. 3d) and human islets (Fig. 3e), with peak at 24 h. The inhibitory effects of HLA-E on immune cells require its expression on the cell surface or its secretion[26]. Flow cytometry confirmed that IFNα increases surface HLA-E expression (Fig. 3f, g, Supplementary Fig. 5l), but there was no HLA-E release to the supernatant (Supplementary Fig. 5m). HLA-E mRNA expression was upregulated by 8-fold in human islets of donors with recent-onset T1D in the DiViD study[27] and HLA-E protein expression was significantly increased in insulin-containing islets, but not in insulin-deficient islets, of T1D individuals in comparison to healthy individuals (Fig. 3h, i). HLA-E expression was present in both beta and alpha cells (but not delta cells; Supplementary Fig. 5n) in the islets of people with T1D, with a predominance of expression among

alpha cells as compared to beta cells (Fig. 3j). This may help to explain why alpha cells are more resistant to the immune assault in T1D.

**mRNA and protein modules regulated by interferon-α.** We integrated the RNA-seq and proteomics data (using all the samples from both 8 and 24 h) using the weighted correlation network analysis package (WGCNA)[28]. The heatmaps of the topological overlap matrix from each dataset with module assignment are shown in Fig. 4a. There were initially 32 eigengene modules of mRNAs and 27 of proteins, which were merged (considering a dissimilarity threshold of 0.25) reducing the numbers of mRNA and protein modules to 8 and 7, respectively (Supplementary Fig. 7a–c). The quality of these modules was determined using a combined score of density and separability measures (Methods)[29], which indicated that they were well-defined ($Z_{summary} > 10$) (Supplementary Fig. 7d). WGCNA analysis of the RNA-seq of human islets exposed to IFNα identified well-defined modules of mRNAs (Supplementary Fig. 8a–d), similar to the ones identified in EndoC-βH1 cells exposed to the cytokine (Supplementary Fig. 8e). To focus on central modules induced by IFNα exposure, we selected only the differentially expressed genes (DEG) (Supplementary Data 3) and abundant proteins (DAP) (Supplementary Data 5) in each eigengene module, representing 49% of the protein-coding DEGs and 89% of the DAPs, and then examined the overlap between these datasets. There was a significant overlap between five modules of mRNAs and proteins (minimum of 10 elements in common, FDR < 0.05) (Fig. 4b). The two main new modules, called #1 and #2 (Fig. 4c), were composed of highly correlated mRNAs and proteins (Supplementary Fig. 7e, g) predominantly upregulated by IFNα at both 8 and 24 h (Supplementary Fig. 7f, h). Module #5 also had significantly correlated members (Supplementary Fig. 7i), but enriched in downregulated mRNAs/proteins at both 8 and 24 h (Supplementary Fig. 7j). Interestingly, there was significant enrichment of ATAC-seq gained OCRs in module #2 (Fig. 4d). They were enriched for TF binding motifs including both the pro-inflammatory motifs ISRE / IRF and the islet-specific transcription factor FOXA2 (Fig. 4e).

To identify the gene regulatory network (GRN) of module #2, we integrated information from two sources: (1) literature-based collection of TF-target interactions[30], and (2) the present de novo TF binding motifs and their predicted targets (Supplementary Fig. 9a). This allowed us to add information from cis-regulatory elements (in orange) acting on the IFNα-induced GRN in human beta cells (Supplementary Fig. 9b). A similar approach was used for modules #1 and #5, but considering only data from the literature (Supplementary Fig. 10a, c).

The PPI network InWeb InBio Map[31] was used to assess the presence of protein–protein interaction (PPI) networks in the different modules. This generated networks of interacting proteins for modules #1, #2 and #5 (Fig. 4f and Supplementary Fig. 10b, d) and allowed the recognition of protein communities (grouped by colors) that regulate specific and common biological functions (Fig. 4f and Supplementary Fig. 10b, d). Module #2, which presents the higher number of connections, showed an enrichment for several key biological processes activated by IFNα and relevant for the pathogenesis of T1D, including cellular response to viruses, antigen processing and presentation via MHC class I, inflammatory and acute phase responses (Fig. 4g).

**Interferon-α changes the alternative splicing landscape.** The present high-coverage RNA-sequencing (>200 million reads) allowed the detection of ~47,000 splicing variants, with IFNα-induced 343 differentially expressed transcripts (DETs) at 2 h,

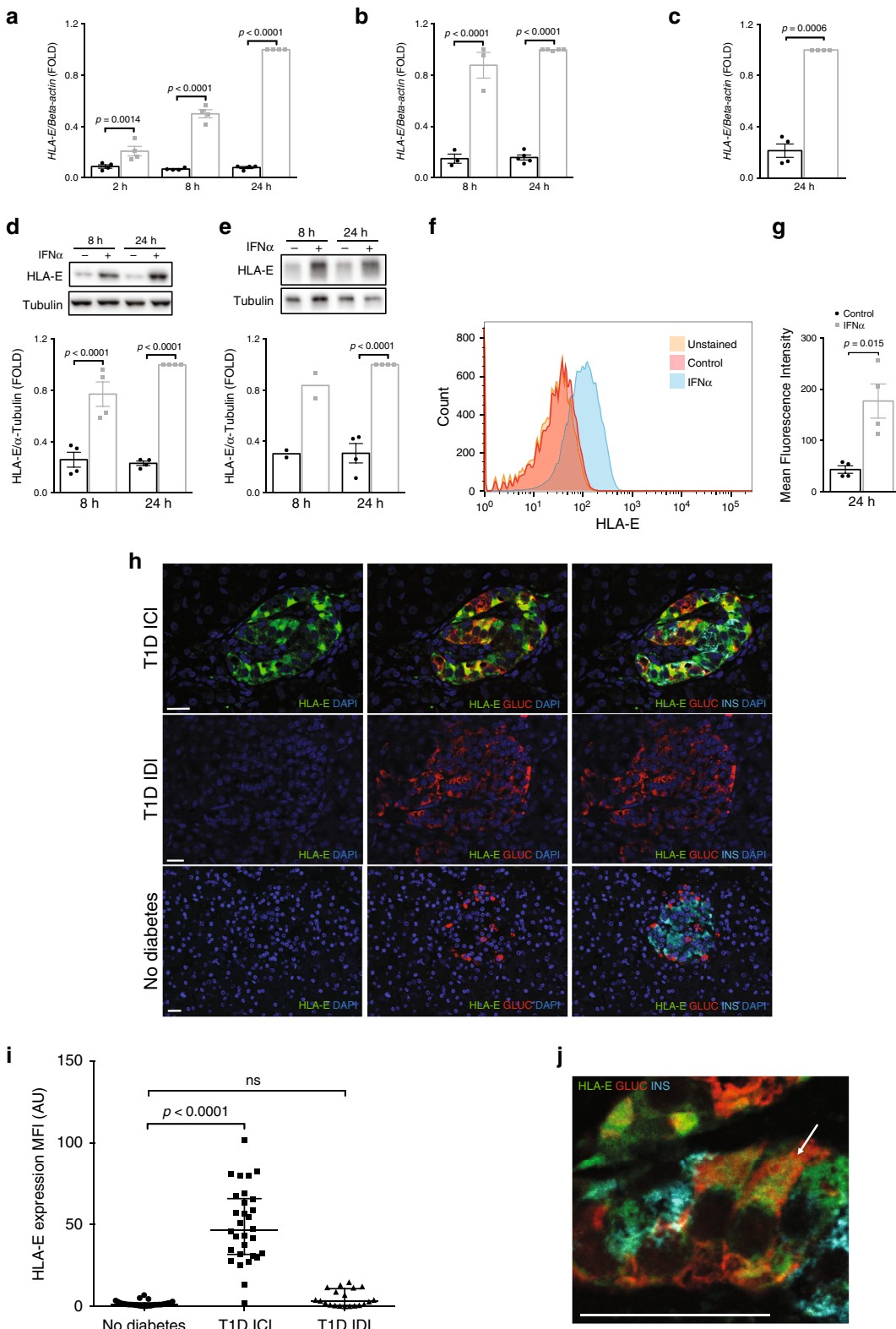

and 1690 and 1669, respectively at 8 and 24 h, with predominance of upregulated transcripts (Fig. 5a and Supplementary Data 6 and 7). Considering all the DETs, 4%, 32% and 32% were exclusively modified at 2, 8 and 24 h, respectively, indicating a predominance of intermediary to late transcriptional changes induced by IFNα. Next, we evaluated the frequency of each individual splicing events (with an absolute difference in percent spliced-in ($|\Delta PSI|$) > 0.2) regulated by IFNα at 8 and 24 h. There were 3140 events at

8 h and 2344 events at 24 h (FDR < 0.05) (Fig. 5b). The most frequent AS event modified by IFNα was cassette exons (CEx), with predominantly increased exon inclusion (represented by $\Delta PSI$ > 0.2, FDR < 0.05) (Fig. 5c). An example of a cassette exon showing increased inclusion upon IFNα treatment is the gene *OASL* (Fig. 5d, e), an antiviral factor targeting single-stranded RNA viruses such as picornaviruses[32]. Exposure to IFNα for 24 h increased exon 4 inclusion in both EndoC-βH1 cells and human

**Fig. 3 HLA-E is overexpressed in pancreatic islets of T1D individuals.** EndoC-βH1 cells (**a**, **d**), human islets (**b**, **e**) or FACS-purified human beta cells (**c**) were exposed (gray bars) or not (black bars) to IFNα for the indicated time points and HLA-E mRNA (**a–c**) and protein (**d**, **e**) evaluated. The values were normalized by the housekeeping gene β-actin (mRNA) or α-tubulin (protein) and then by the highest value of each experiment considered as 1 (for **a** ($n =$ 4); **b** ($n = 3$ (8 h), $n = 5$ (24 h)); **c** ($n = 4$); **d** ($n = 4$) and **e** ($n = 2$ (8 h), $n = 4$ (24 h)), ANOVA with Bonferroni correction for multiple comparisons (**a–e**)). **f**, **g** HLA-E cell surface expression was quantified in EndoC-βH1 cells by flow cytometry. Histograms (**f**) represent changes in mean fluorescence intensity (MFI). The MFI values (**g**) were quantified at baseline and after 24 h exposure to IFNα ($n = 4$, two-sided paired *t*-test). Values are mean ± SEM (**a–g**). **h** Immunostaining of HLA-E (green), glucagon (red) and insulin (light blue) in representative islets from individuals with or without diabetes. The top and middle panels represent an insulin-containing islet (ICI) and insulin-deficient islet (IDI) from T1D sample DiViD 3, and the lower panel represents an islet from a control donor (EADB sample 333/66). DAPI (dark blue). Scale bar 20 μm. **i** The MFI analysis of HLA-E expression. 30 ICIs from 6 independent individuals with T1D (5 islets per individual), 20 IDIs from 4 independent individuals with T1D (5 islets per individual), and 30 ICIs from 6 independent individuals without diabetes (5 islets per individual) were analyzed. Values are median ± interquartile range; ANOVA with Bonferroni correction for multiple comparisons, AU (arbitrary units), ns = (non-significant). **j** Higher magnification image demonstrating that HLA-E (green) localizes predominantly to alpha cells in a T1D donor islet (glucagon (red); insulin (light blue)) but is also expressed in beta cells, as indicated in **h** and **j**. Scale bar 30 μm. Source data are provided as Source Data file.

islets (Fig. 5d). In line with this, the protein encoded by the isoform *OASL−001* (which retains exon 4) displayed a higher IFNα-induced upregulation in comparison with the protein encoded by the isoform *OASL−002*, which has exon 4 exclusion (Fig. 5e). Interestingly, the isoform *OASL−001* has antiviral activity, whereas the isoform 002 lacks the ubiquitin-like domain required for this response (Supplementary Fig. 11A)[33].

Intron retention is an important mechanism of gene expression regulation, promoting nuclear sequestration of transcripts or cytoplasmatic degradation via nonsense-mediated decay[34]. There was a predominance for intron removal after 24 h (represented by ΔPSI < −0.2, FDR < 0.05), but not at 8 h (Fig. 5f). To understand how this impacts protein translation, we compared changes in protein abundance among three categories of ΔPSI. Genes presenting intron removal had a significant increase in protein expression after IFNα exposure for 24 h in comparison to those with intron retention (ΔPSI > 0.2, FDR < 0.05) or with non-significant intron changes (ΔPSI −0.2–0.2 or FDR > 0.05) (Fig. 5g).

There were clear variations in the mRNA expression of several well-known RNA-binding proteins (RBPs)[35] upon IFNα exposure (Fig. 5h, left panel), but the impact on the respective proteins was less pronounced (Fig. 5h, right panel). We focused on a group of IFNα-modified RBPs at both mRNA and protein levels after 24 h, and mapped their RNA-binding motifs among upregulated and downregulated alternative exons. In support of a biological role for these RBPs on alternative exon splicing, there was an enrichment of their binding motifs in regions controlling alternative cassette exon inclusion/exclusion (Fig. 5i). To further study some of these findings, we first reproduced the IFNα-induced downregulation of two RBPs, ELAV-like protein 1 (*ELAVL1*) and heterogeneous nuclear ribonucleoprotein (*HNRNPA1*), by using specifics siRNAs (Supplementary Fig. 12a, e). Next, we evaluated whether this inhibition reproduced the changes induced by IFNα in the exon usage of four-and-a-half LIM domain protein 1 (*FHL1*) and Caprin Family Member 2 (*CAPRIN2*) (Supplementary Fig. 12b, f) two potential targets of, respectively, *ELAVL1*[36] and *HNRNPA1*[37]. Silencing these RBPs promoted changes on exon usage (Supplementary Fig. 12c, g) that were similar to the ones observed after IFNα treatment (Supplementary Fig. 12b, f). This is especially relevant in the context of the IFNα-induced exon exclusion FHL1, which decreases the expression of transcripts coding for the protein FHL1A (Supplementary Fig. 12d), an isoform described as a key host factor for the replication of the RNA virus Chikungunya[38].

RBPs can also control gene expression by blocking RNA translation, as described for the Fragile X Mental Retardation 1 (*FMR1*) gene[39]. Indeed, there was a significant downregulation of previously validated bona fide targets of *FMR1* (Supplementary Table 1)[40] in IFNα-treated EndoC-βH1 cells as compared to the remaining proteins (Fig. 5j).

**IFNα induces increased alternative transcription start sites.** The usage of alternative transcription start (TSS) sites is another mechanism that generates different transcripts from the same gene[41]. We used the SEASTAR pipeline[42] for the computational identification and quantitative analysis of first exon usage. This approach recognized >250 events of alternative first exon (AFE) usage occurring in 166 different genes at 8 h, and >130 events of AFE usage in 88 genes at 24 h (Fig. 6a). In agreement with this, 118 and 64 alternative promoters (±2 kb around FE TSS) detected by SEASTAR at 8 and 24 h, respectively, overlapped peaks of TSS identified by the FAMTOM5 Consortium[43]. Among these genes was the 5′-nucleotidase cytosolic IIIA (*NT5C3A*), a negative regulator of IFN-I signaling[44]. This gene had two AFEs identified by the SEASTAR modeling. In untreated condition (controls), there was a higher usage of the proximal first exon (FE), present in the isoforms *NT5C3A−001* and 002 in beta cells (Fig. 6b, upper panel). After INFα exposure, however, there was increased usage of the distal FE from the transcript *NT5C3A−004* (ΔPSI: 0.71 (8 h)/0.65 (24 h), both FDR < 0.001), which is supported by the cap analysis of gene expression (CAGE) of TSSs[45] (Fig. 6b, upper panel). This was confirmed in independent samples of EndoC-βH1 cells and human islets using specific primers (Fig. 6b, lower panel). Exon Ontology analysis[46] indicated that this FE shift probably has functional impact, since the distal FE lacks both the endoplasmic reticulum (ER) retention signal and the transmembrane helix (Supplementary Fig. 11b), enabling its encoded protein to remain in the cytosol where *NT5C3A* acts[44].

Next, we compared the frequency of gained OCRs among alternative promoters. As the SEASTAR pipeline mainly recognizes non-redundant FEs, we evaluated alternative promoters identified by both the SEASTAR pipeline and the FAMTON5 database of alternative TSSs[45] (Supplementary Methods). We thus identified 198 and 51 gained OCRs present in alternative promoter regions at 2 and 24 h, respectively. Characterization of the IFNα-induced alternative promoters presenting a major gain in chromatin accessibility pointed to the T1D risk gene *RMI2*[47]. At gene level, there was only a ~1.4-fold upregulation of *RMI2* expression, but at the transcript level there was a >60-fold increase in two isoforms, *RMI2−002* and −004. Visualization of the *RMI2* locus combined with ATAC-seq and RNA-seq peaks indicated that the isoform *RMI2−004* gained chromatin accessibility in its promoter leading then an increase in mRNA expression (Fig. 6c). Data from CAGE analysis[45] and RNA polymerase II ChIP-seq of another human cell type exposed to IFNα[48] (Fig. 6c, lower part) confirms the presence of the *RMI2*

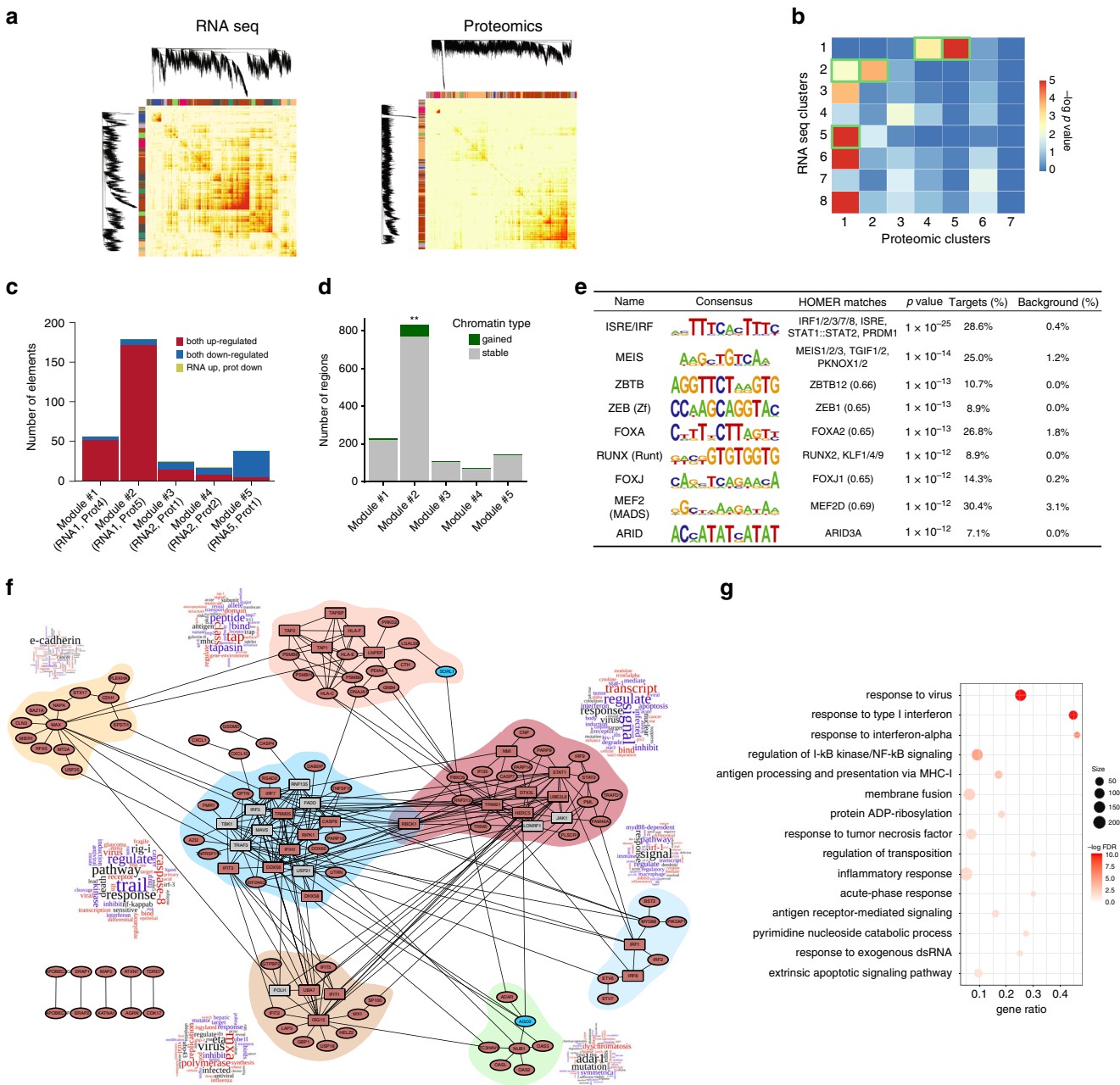

**Fig. 4 Weighted correlation network analysis (WGCNA) identifies IFNα-regulated mRNA and protein modules. a** Heatmap representation of the topological overlap matrix. Rows and columns correspond to single genes/proteins, light colors represent low topological overlap, and progressively darker colors represent higher topological overlap. The corresponding gene dendrograms and initial module assignment are also displayed. **b** Identification of modules presenting significant overlap (FDR < 0.05 and a minimum of 10 members in common) (green border) between differentially expressed genes (DEG) and their translated differentially abundant proteins (DAP). **c** Composition, number of elements and type of DEG and DAP present in each of the significantly overlapping modules. **d** ATAC-seq-identified open chromatin regions at 2 h were linked to gene transcription start sites (TSSs) in a 40 kb window. These genes and their open chromatin regions were associated to the modules of DEG and DAP. The enrichment for gained open chromatin regions was then evaluated in each module. (** represents a *p*-value = 0.002343, one-sided $\chi^2$ test). **e** De novo HOMER motifs present in the ATAC-seq regions overlapping module #2 as described in Methods. The unadjusted *p*-values were obtained using the hypergeometric test from the HOMER package[77]. **f** The protein–protein interaction (PPI) network of module #2 was done using the InWeb InBio Map database[31]. Enriched proteins (FDR < 0.05 and minimum number of connections = 5, represented as squares) were identified and added to the network if they were not already present. Red fill identifies upregulated proteins, blue fills downregulated proteins and gray fill equal-regulated. Colored regions delimitate communities of proteins, as described in Methods. The wordcloud next to each community presents their enriched geneRIFs terms. **g** The biological processes (GO) overrepresented in module #2 summarize the main findings observed in IFNα-treated human beta cells. The present results were based on RNA-seq (*n* = 5) and proteomics (*n* = 4) data of EndoC-βH1 cells.

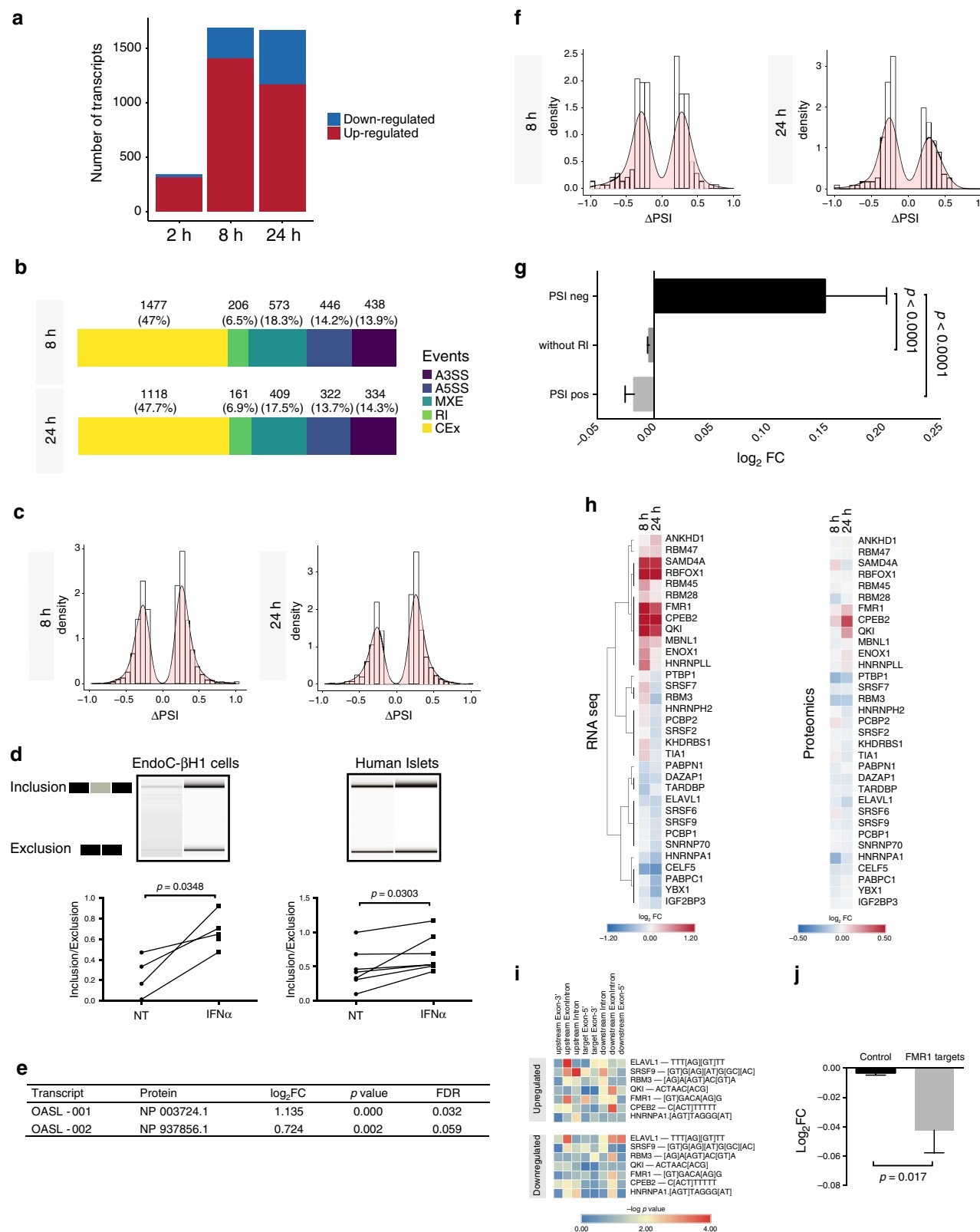

alternative promoter. The IFNα-induced *RMI2*−004 upregulation was confirmed using specific primers in both EndoC-βH1 cells and human islets (independent samples) (Fig. 6d). These findings support a double mechanism by which IFNα affects human beta cells, i.e. first a massive change in open chromatin regions followed by later changes in gene expression and AS (see above) and also AFE usage.

**Mining IFNα signatures to identify T1D therapeutic targets.** Considering the significant overlap observed between gene profiles of IFNα-exposed EndoC-βH1 cells and beta cells from T1D individuals (Supplementary Fig. 2d), mining these common signatures might identify relevant T1D therapeutic targets. First, the top 150 commonly upregulated genes detected by the RRHO analysis of both IFNα-exposed EndoC-βH1 cells and beta cells

**Fig. 5 Interferon-α changes the alternative splicing landscape. a** EndoC-βH1 cells were exposed to IFNα for the indicated time points. The significantly upregulated (red) and downregulated (blue) transcripts were identified using Flux Capacitor ($n = 5$, $|\log_2 FC| > 0.58$ and FDR < 0.05). **b** Frequency of individual alternative splicing events regulated by IFNα ($n = 5$, $|\Delta PSI| > 0.2$, minimum 5 reads, FDR < 0.05). **c** Frequency distribution of alternative cassette exon (CEx) events altered by IFNα (($n = 5$, $\Delta PSI) > |0.2|$ and FDR < 0.05). **d** Confirmation of the increased exon 4 inclusion in the antiviral gene OASL by IFNα (24 h). cDNA was amplified by RT-PCR using primers located in the upstream and downstream exons of the splicing event and the product evaluated using a Bioanalyzer 2100 ($n = 4$ (EndoC) and $n = 7$ (human islets), two-sided paired $t$-test). **e** The $\log_2 FC$s of the proteins coding for OASL-001 and −002 isoforms from IFNα-treated EndoC-βH1 cells proteomics (24 h) ($n = 4$). **f** Frequency distribution of retained intron (RI) events altered by IFNα ($n = 5$, $|\Delta PSI| > 0.2$ and FDR < 0.05). **g** The protein $\log_2 FC$ values obtained by proteomics analysis of EndoC-βH1 cells exposed to IFNα for 24 h were classified in three categories according to the levels of retained intron $\Delta PSI$ ($n = 5$, mean ± SEM, ANOVA with Bonferroni correction). **h** Expression of RNA-binding proteins (left) that are significantly modified at mRNA level (FDR < 0.05) after exposure to IFNα and their respective proteins (right) in the indicated time points ($n = 4$–5). **i** Positional enrichment of motifs from significantly modified RBPs among regions involved in the regulation of modified cassette exons (CEx) after exposure to IFNα for 24 h. ($n = 5$, $|\Delta PSI| > 0.2$, FDR < 0.05). **j** Comparison between the $\log_2 FC$ of a curated list (Supplementary Table 1) of known FMR1 target proteins against the $\log_2 FC$ of the remaining proteins detected by the proteomics of EndoC-βH1 cells exposed to IFNα for 24 h ($n = 4$, mean ± SEM; two-sided unpaired $t$-test). Source data are provided as a Source Data file.

from T1D individuals were selected (Supplementary Fig. 2d and Fig. 7a) to query the Connectivity Map database[49]. We focused in opposite signatures of perturbagens that may reverse the effects of IFNα. To decrease off-target findings based on individual compounds, the analysis was performed considering only the classes of perturbagens. Four main classes, including bromodomain inhibitors, potentially reversed the signature from our query (tau score < −90) (Fig. 7b). Comparable results were obtained when analyzing the intersection of IFNα-exposed pancreatic human islets and beta cells from T1D individuals (Supplementary Fig. 13a). Bromodomain inhibitors have been shown to prevent autoimmune diabetes in animal models[50] and the KD of the bromodomain containing 2 gene (BRD2) induced an opposite signature to our model (Supplementary Fig. 13b). Pre-treatment of EndoC-βH1 cells with two bromodomain inhibitors decreased both IFNα-induced HLA-I and CXCL10 induction, with no changes in CHOP (DDIT3) expression (Fig. 7c, e) or in apoptosis induced by IL1β + IFNα (Fig. 7d, f). In human islets, these inhibitors induced a ~30% decrease in IFNα-induced HLA-I expression and a 90% reduction in CXCL10 expression; at least in the context of I-BET-151, there was a 60% reduction of the ER stress marker CHOP (DDIT3) (Supplementary Fig. 13c, d).

Next, we searched for clinically approved drugs (DrugBank 5.1[51]) among the PPI network of the WGCNA module #2 (Fig. 4f), with a view to possible drug repurposing. Module #2 is particularly interesting in this context as it recapitulates many of the key biological processes induced by IFNα (Fig. 4g), and because ~50% of its members were also present among the most upregulated genes from the RRHO analysis (Supplementary Fig. 2d). An interesting target recognized as a hub for different drugs was the kinase JAK1 (Fig. 8a) and its inhibitor baricitinib, which has shown promising effects in the treatment of human rheumatoid arthritis[52]. Baricitinib prevented IFNα-induced mRNA expression of HLA-I, CXCL10 and CHOP (DDIT3) in EndoC-βH1 cells (Fig. 8b) and human islets (Fig. 8c) and it completely protected EndoC-βH1 cells (Fig. 8d) and human islets (Fig. 8e) against the pro-apoptotic effects of IFNα + IL1β. Furthermore, baricitinib decreased the cell surface protein expression of MHC class I by >90% in EndoC-βH1 cells (Fig. 9a) and human islets (Fig. 9b, c).

## Discussion

We presently modeled the initial changes observed in the islets of Langerhans during T1D by performing an integrated multi-omics approach in EndoC-βH1 cells exposed to the early cytokine IFNα. The model was validated using human islets RNA-seq and independent experiments using the same human beta cell line, pancreatic human islets and FACS-purified human beta cells. Of relevance, taking into account the major differences between

human and rodent beta cell responses to stressful stimuli[53,54], all experiments were performed in clonal or primary human beta cells/islets. This approach identified very rapid and broad beta cell responses to IFNα including: (1) major early modifications in chromatin remodeling, which activates regulatory elements; (2) the key TFs regulating signaling, and the crosstalk between beta cells and immune cells; (3) the functional modules of genes and their regulatory networks; and (4) alternative splicing and first exon usage as important drivers of transcript diversity. Finally, an integrative analysis led to the identification of two compound classes that reverse all or part of these alterations in EndoC-βH1 cells and human islets and may be potential therapeutic targets for future trials in T1D prevention/treatment.

During viral infection a prompt innate immune response, mediated to a largest extent via type I interferons, is critical to control virus replication and spreading[55]. In line with this, exposure of human beta cells to IFNα leads to changes in chromatin accessibility already at 2 h, which correlates with subsequent changes in mRNA and protein expression at 8 and 24 h. The majority of these regions are localized distally to TSSs, indicating that they may act primarily as distal regulatory elements. Interestingly, these regions were enriched in motifs of islets-specific TFs, suggesting that tissue-restricted characteristics regulate the local responses during insulitis, as we have recently described for the cytokines IL1β + IFNγ[19]. This could explain the preferential expression of HLA class I (both the classical ABC members and the presently described inhibitory HLA-E) by pancreatic islets in comparison to the surrounding exocrine pancreas. Islet HLA class I overexpression is a key finding during T1D development[56], contributing for the recruitment of autoreactive CD8+ T cells that selectively attack beta cells[1]. IFNα also induces pathways involved in protein modification (ubiquitination, sumoylation, etc), degradation (proteasome, etc) and ER stress, which can generate neoantigens[14].

The IRF and STAT family members are master TFs involved in IFN-I signaling[2]. Viruses have developed several species-specific mechanisms to antagonize STAT1 and STAT2 activation[55]. For instances, the NS5 protein of Zika virus degrades human but not mouse STAT2[57]. In the present work, we confirmed the importance of both STAT1 and 2 for INFα signaling in beta cells, and observed that their individual KD is compensated in most cases by the remaining member, as a possible backup mechanism to protect against pathogens[58]. Interestingly, IRF1 seems to be a critical regulator of the IFNα-mediated "defense" responses in beta cells, including induction of checkpoint proteins such as PDL1 and HLA-E (present data), and the suppressors of cytokine signaling 1 and 3 (SOCS3) (ref. [59] and present data). This stands in contrast to its pro-inflammatory effects in immune cells[60]. In line with the possible role for IRF1 in dampening islet

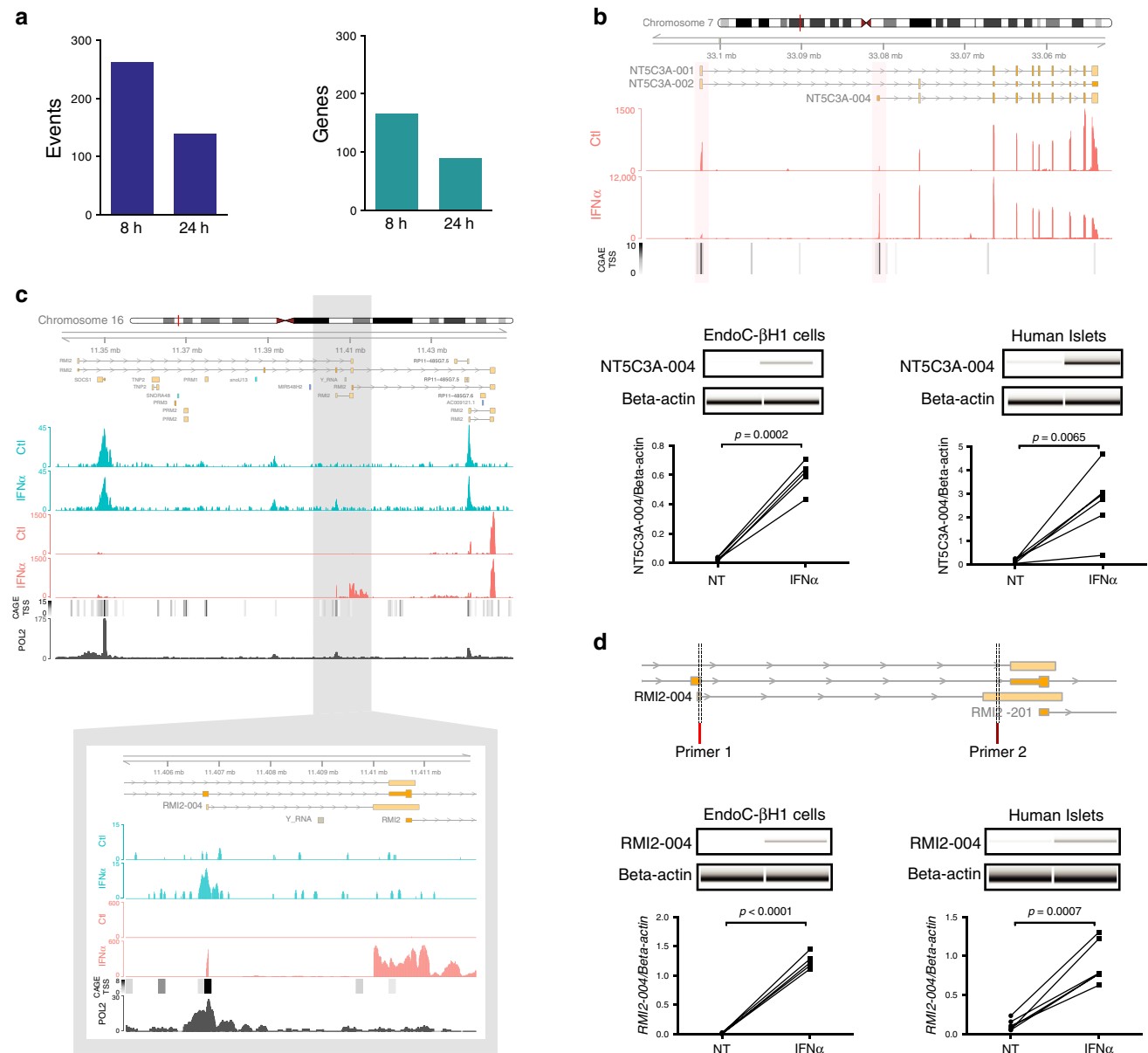

**Fig. 6 Changes in the alternative transcription start site (TSS) initiation increase the repertoire of IFNα-regulated transcripts. a** The tool SEASTAR[42] was used to estimate the frequency of differential alternative first exon (AFE) usage induced by IFNα in human beta cells. The total number of IFNα-dependent AFEs events (left) and number of genes with AFEs (right) in the indicated time point are shown (n = 5, ΔPSI > |0.2|, FDR < 0.05). **b** View of the NT5C3A locus showing the transcripts with AFE usage, the RNA-seq (red) signals of EndoC-βH1 cells exposed or not to IFNα and the CAGE TSSs information (black scale)[45] (upper panel). Confirmation of the AFE usage identified by SEASTAR in the gene NT5C3A (lower panel). cDNA was amplified by RT-PCR using primers located in the AFE and in its downstream exon. The PCR products were analyzed by automated electrophoresis using a Bioanalyzer 2100 and quantified by comparison with a loading control. The values were then corrected by the housekeeping gene β-actin. (n = 4 (EndoC) and n = 6 (human islets), two-sided paired t-test). **c** View of the RMI2 locus showing all the transcripts in this region, the ATAC-seq (blue) and the RNA-seq (red) signals of EndoC-βH1 cells exposed or not to IFNα for 24 h, the CAGE TSSs information (black scale)[45] and RNA polymerase II ChIP-seq signal of human K562 cells exposed to IFNα (black)[48]. A higher magnification of the RMI2-004 locus is presented below (image representative of 4–5 independent experiments). **d** Confirmation of the AFE usage in the gene RMI2. Genome mapping (upper part) showing the genomic regions used to design-specific primers located in the AFE of the transcript RMI2-004 and in its downstream exon. The PCR product was analyzed by automated electrophoresis using a Bioanalyzer 2100 and quantified by comparison with a loading control. The values were then corrected by the housekeeping gene β-actin. (n = 4 (EndoC) and n = 6 (human islets), two-sided paired t-test). Source data are provided as a Source Data file.

inflammation, systemic knockout of IRF1 prevents autoimmune diabetes in NOD mice[61], whereas IRF1 deletion in islets is associated with shorter mouse allograft graft function and survival[62].

Alternative splicing (AS) is a species, tissue and context-specific post-transcriptional mechanism that expands the number of transcripts originated from the same gene thus increasing protein diversity[63]. Pancreatic beta cells share many characteristics with neuronal cells, including analogous signal transduction, developmental steps and splicing networks[64]. Both T1D risk genes[65] and the cytokines IL1β + IFNγ[10] modify AS in beta cells. We presently identified a preferential alternative exon inclusion after IFNα exposure and mapped the potentially involved RBPs, which included the upregulated protein Quaking (QKI). QKI activation

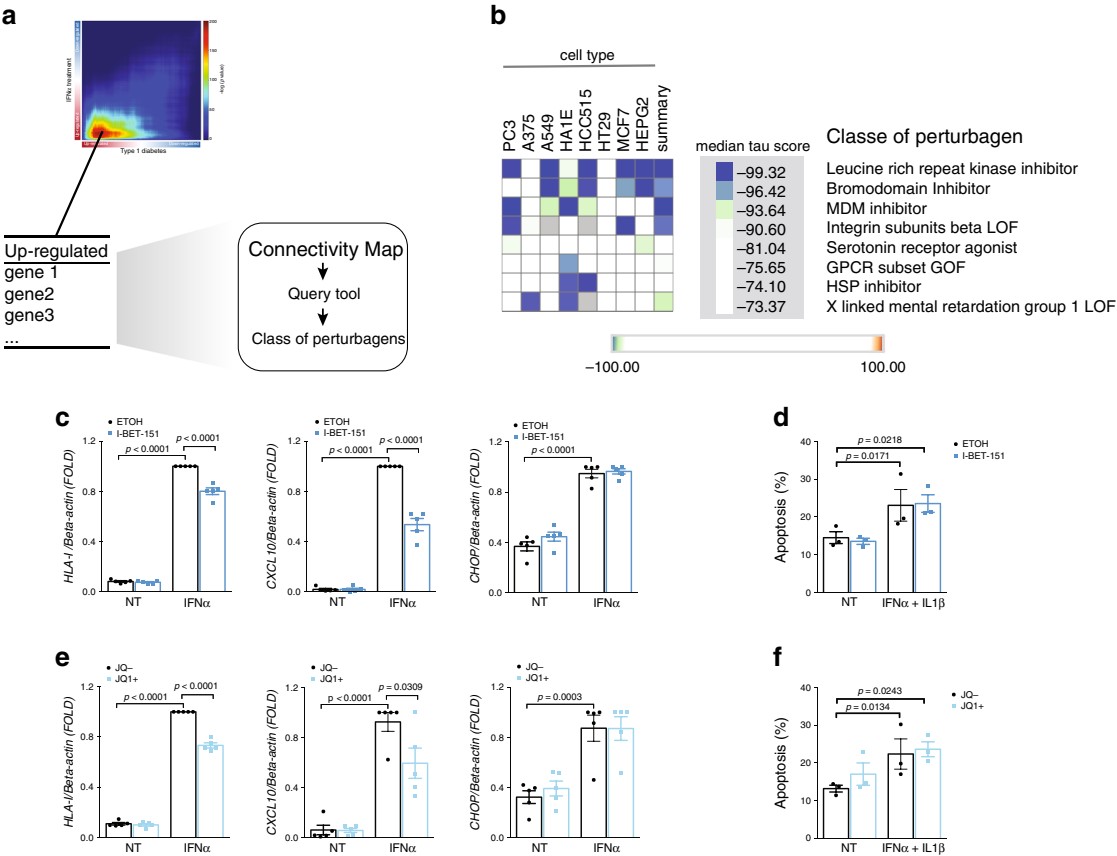

**Fig. 7 Mining the type I interferon signature of pancreatic beta cells for identification of potentially T1D therapeutic targets. a** The top 150 upregulated genes identified in Supplementary Fig. 2d were used to query the Connectivity MAP database of cellular signatures[49]. **b** Connectivity map classes of perturbagens that promote an opposite signature to the one shared between beta cells of T1D individuals and EndoC-βH1 cells exposed to IFNα (Supplementary Fig. 2d). **c, e** EndoC-βH1 cells were pretreated for 2 h with the bromodomain inhibitors I-BET-151 (1 μM) (**c**) or JQ1+ (0.4 μM) (**e**) and then exposed to IFNα for 24 h. Cells were collected and the mRNA expression for HLA class I (ABC), the chemokine CXCL10 and the ER stress marker CHOP (DDIT3) evaluated. Ethanol (vehicle) and an inactive enantiomer (JQ1−) were used as respective controls for I-BET-151 and JQ1+. (n = 5, mean ± SEM, ANOVA with Bonferroni correction for multiple comparisons). **d, f** Cell viability after exposure to the combination of cytokines IFNα (2000 U/ml) + IL1β (50 U/ml) in the presence or not of the bromodomain inhibitors (n = 3, mean ± SEM, ANOVA with Bonferroni correction for multiple comparisons).

in monocytes promotes extensive changes in AS, favoring their differentiation into pro-inflammatory macrophages[66]. Furthermore, QKI binds to the genome of RNA viruses and inhibits their replication[67]. A similar mechanism was recently described for *FMR1*[68], another RBP induced by IFNα, which controls protein translation in beta cell (present data). Several other RBPs were observed as downregulated by IFNα and identified as potential regulators of IFNα-induced AS events. Thus, inhibition of ELAVL1 and HNRNPA1 reproduced IFNα-mediated changes in exon usage. Different RNA viruses can use both ELAVL1[69] and HNRNPA1[70] to support their replication, indicating that the decreased expression of these proteins may provide an additional IFN-triggered antiviral mechanism. These findings suggest that during potentially diabetogenic viral infections, RBPs may have a dual role: first as splicing regulators and second as regulators of viral replication.

In order to identify novel approaches to protect beta cells in T1D, we analyzed the similarities between beta cell signatures from T1D donors and those following IFNα exposure, and compared the top identified genes/pathways with the Connectivity Map[49] and the DrugBank[51] database. This identified two groups of potential therapeutic agents, namely bromodomain and JAK inhibitors. Bromodomain (BRD) proteins are components of chromatin-remodeling complexes that promote chromatin decompaction and transcriptional activation. BET inhibitors have

shown protective effects in different animal models of autoimmunity[71], including the diabetes-prone NOD mice[50]. We have now expanded these findings to human beta cells, showing that two distinctive BET inhibitors (JQ1+ and I-BET-151) decrease IFNα-induced responses, including HLA class I and chemokine overexpression.

After binding to its receptor, IFNα promotes phosphorylation of two tyrosine kinases, JAK1 and TYK2, which then trigger the downstream signaling cascade. Chemical inhibition of JAK1 + JAK2 prevents autoimmune diabetes in NOD mice[72] and polymorphisms associated with decreased TYK2 function are protective against human T1D[73]. We presently observed that baricitinib, a JAK1/2 inhibitor recently approved for use in rheumatoid arthritis by the FDA[52], decreased all the three hallmarks previously identified in islets of T1D individuals and initiated by IFNα in human beta cells, namely HLA class I overexpression, ER stress and beta cell apoptosis, supporting its future testing in T1D.

In conclusion, we have applied a multi-omics approach to study the different levels of gene regulation induced by IFNα in EndoC-βH1 cells and pancreatic human islets. This in vitro modeling showed strong correlation with the mRNA profile from beta cells of T1D individuals. At the genomic level, early chromatin remodeling activated *cis*-regulatory elements, many of them presenting motifs for islets-specific TFs, providing a

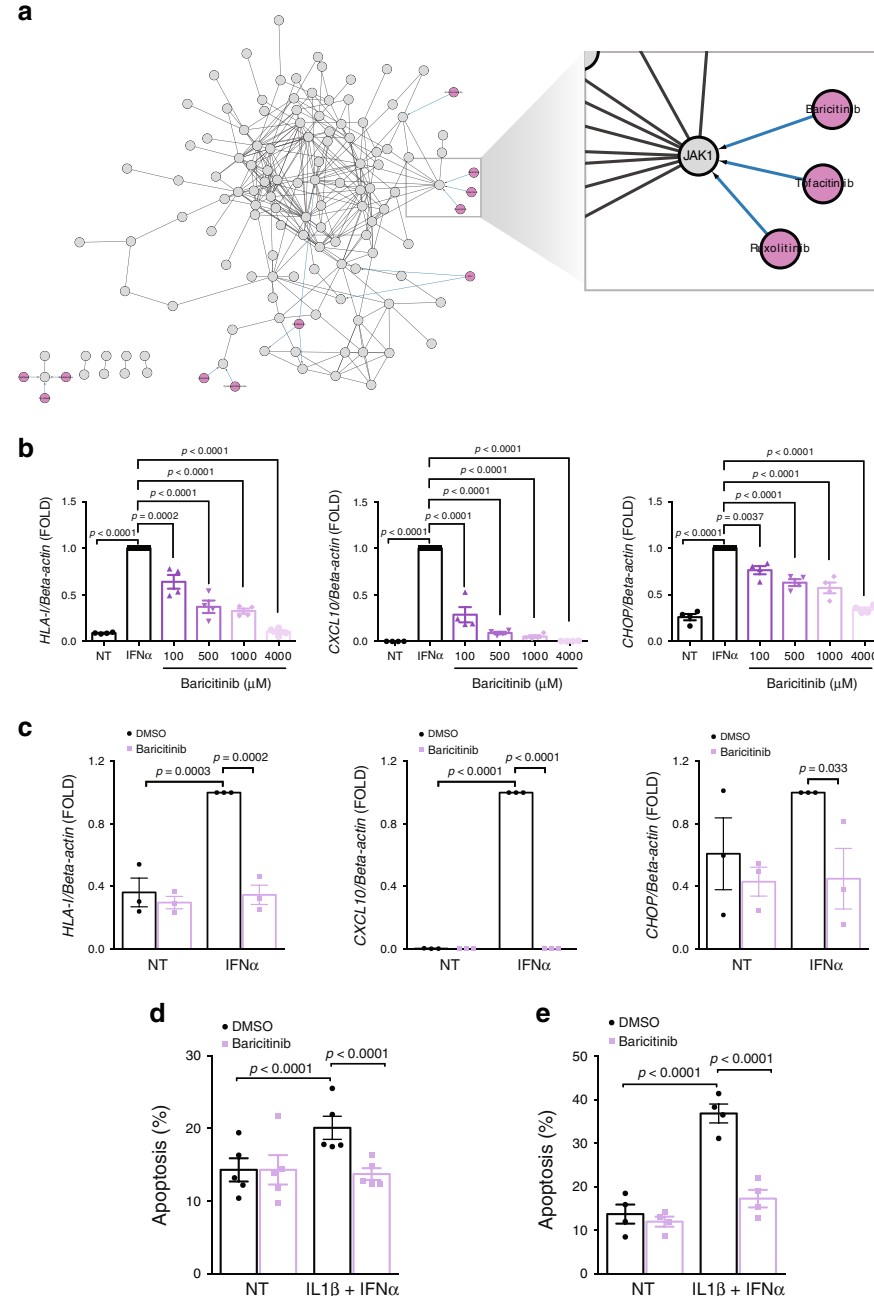

**Fig. 8 Establishing JAK1 inhibition as protective mechanism against IFNα-mediated inflammation and apoptosis. a** The PPI network of module #2 was integrated with the DrugBank repository[51] using the CyTargetLinker app[78] in Cytoscape. A higher magnification on JAK1 is shown. **b** EndoC-βH1 cells were pretreated with DMSO (NT) or baricitinib at the indicated concentrations for 2 h. Cells were then left untreated (black bars), or treated with IFNα alone (white bars) without or with the presence of different concentrations of baricitinib (purple scale bars) for 24 h and mRNA expression of HLA class I (ABC), CXCL10 and CHOP (DDIT3) analyzed. The values were normalized by the housekeeping gene β-actin and then by the highest value of each experiment considered as 1 ($n = 4$, mean ± SEM, ANOVA with Bonferroni correction for multiple comparisons). **c** Human islets were pretreated with baricitinib (4 μM) or DMSO (vehicle) and then exposed or not to IFNα for 24 h in the presence or not of baricitinib. mRNA expression of HLA class I (ABC), CXCL10 and CHOP (DDIT3) was analyzed and values normalized by the housekeeping gene β-actin and then by the highest value of each experiment considered as 1. ($n = 3$, mean ± SEM, ANOVA with Bonferroni correction for multiple comparisons). **d**, **e** EndoC-βH1 cells (**d**) and human islets (**e**) were pretreated with DMSO or baricitinib (4 μM) for 2 h. Subsequently, cells were left untreated or treated with IFNα (2000 U/ml) + IL1β (50 U/ml) in the absence or presence of baricitinib for 24 h. Cell viability was evaluated using nuclear dyes by two independent observers. (**d** ($n = 5$), **e** ($n = 4$), mean ± SEM, ANOVA with Bonferroni correction for multiple comparisons). Source data are provided as a Source Data file.

possible mechanism by which tissue-restricted autoimmune diseases might arise. Post-translational modifications, alternative splicing and first exon usage were induced by IFNα, likely expanding the repertoire of proteins and transcripts generated by beta cells in response to this inflammatory stimuli. This can also be a source of potential neoantigens. Interestingly, IFNα-exposed human beta cells upregulate co-inhibitory proteins such as PDL1 and HLA-E, which may attenuate or delay the autoimmune assault. Finally, the present results provide a useful resource for the discovery of compounds that may be used to reverse the

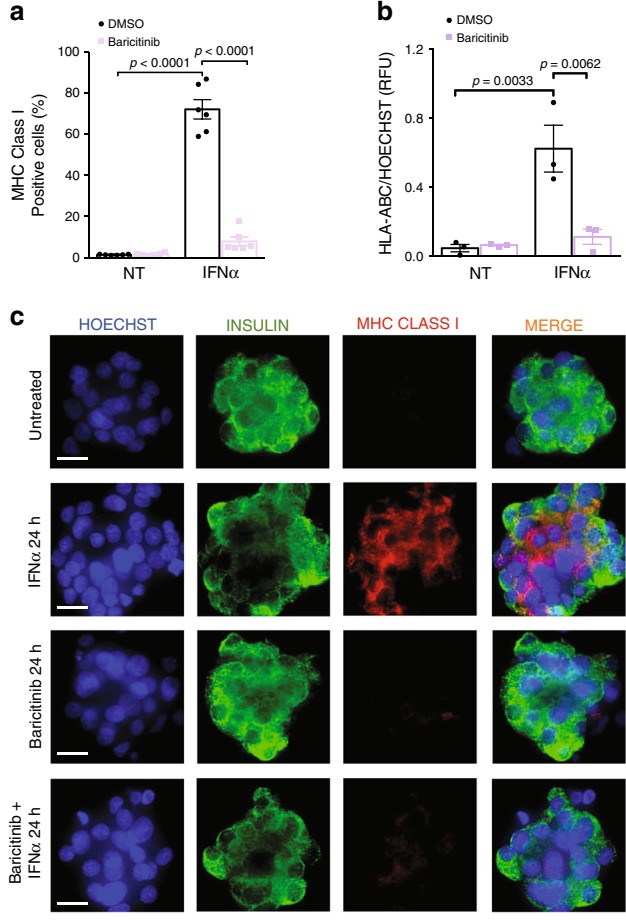

**Fig. 9 Baricitinib decreases IFNα-mediated MHC class I protein expression in beta cells. a** EndoC-βH1 cells were pretreated with baricitinib (4 μM) or DMSO and then exposed or not to IFNα for 24 h in the presence or not of baricitinib. MHC class I (ABC) protein expression was measured by flow cytometry. The percentage of positive cells was quantified. ($n = 6$, mean ± SEM, ANOVA with Bonferroni correction for multiple comparisons). **b**, **c** Dispersed human islets were pretreated with baricitinib (4 μM) or DMSO (vehicle). Next, cells were left untreated, treated with IFNα alone or with IFNα in the presence of baricitinib for 24 h. MHC class I intensity was quantified in each condition (**b**) using Fiji software[80] and normalized by the HOECHST intensity to correct for the number of cell per area ($n = 3$, ANOVA with Bonferroni correction for multiple comparisons, RFU (relative fluorescence units)). Immunocytochemistry (ICC) analysis (**c**) of MHC class I (ABC) (red), insulin (green) and HO (blue) was performed to confirm MHC class I expression in three independent human islet preparations. Scale bar 10 μm.

effects of IFNα on human pancreatic beta cells, paving the way for potential T1D interventional trials.

## Methods

**Culture of EndoC-βH1 cells and human islets, cell treatment**. The human pancreatic beta cell line EndoC-βH1 was kindly provided by Dr. R. Scharfmann, University of Paris, France[74]. Human islet isolation from 20 non-diabetic organ donors (Supplementary Table 2) was performed in accordance with the local Ethical Committee in Pisa, Italy. The use of pancreatic human islets for this project was approved by the Comité d'Ethique hospitalo-facultaire Erasme-ULB. These cells were maintained in culture and treated as described in Supplementary Methods.

**FACS-purified human beta cells isolation and treatment**. Whole pancreatic human islets were exposed or not to IFNα for 24 h. After this period, the islets were dispersed into single cells and surface staining was carried out in FACS buffer (PBS with 0.5% BSA and EDTA 2 mM final concentration). Indirect antibody labeling was performed with two sequential incubation at 4 °C and one wash in FACS buffer followed each step. Cells were resuspended in FACS buffer, viability dye was added (DAPI) and cells were sorted on a FACSAria III cell sorter (BD Biosciences). Primary (mouse anti-human NTPDase3, hN3-B3S, www.ectonucleotidases-ab.com) and secondary (Alexa Fluor 546 conjugated donkey anti-mouse IgG (A10036, Thermo-Fisher Scientific)) antibodies were used with the dilutions described in Supplementary Table 5. Data analysis was carried out with FlowJo software (Version 10).

**ATAC sequencing processing and analysis**. ATAC sequencing was performed in four independent experiments for each time point (2 and 24 h)[75]. For ATAC-seq 50,000 EndoC-βH1 cells were exposed or not to IFNα for 2 or 24 h. After that, the cells were harvested, and the nuclei isolated by using 300 μl of cold lysis buffer (10 mM Tris–HCl pH 7.4, 10 mM NaCl, 3 mM MgCl₂, 0.1% Igepal CA-630). The nuclei pellet was resuspended in a 25 μl transposase reaction mix containing 2 μl of Tn5 transposase per reaction and incubated at 37 °C for 1 h. The tagmented DNA was isolated using SPRI cleanup beads (Agencourt AMPure XP, Beckman Coulter). For library amplification two sequential 9-cycle PCR were performed (72 °C for 5 min; 98 °C for 30 s; 9 cycles of 98 °C for 10 s, 63 °C for 30 s; and 72 °C for 1 min; and at 4 °C hold). Finally, the DNA library was purified using the MinElute PCR Purification Kit (Qiagen, Venlo, Netherlands). TapeStation and semi-quantitative PCR assays at target positive and negative controls were performed to ensure the quality and estimate the efficiency of the experiment before sequencing. Libraries were sequenced single-end on an Illumina HiSeq 2500. Data processing and analysis is described in Supplementary Methods.

**RNA-sequencing processing and analysis**. Total RNA of five independent experiments with EndoC-βH1 cells and six independent preparation of pancreatic human islets exposed or not to IFNα for different time points was obtained using the RNeasy Mini Kit (Qiagen, Venlo, Netherlands). RNA integrity number (RIN) values were evaluated using the 2100 Bioanalyzer System (Agilent Technologies, Wokingham, UK). All the samples analyzed had RIN values >9. mRNA was obtained from 500 ng of total RNA using oligo (dT)beads, before it was fragmented and randomly primed for reverse transcription followed by second-strand synthesis to generate double-stranded cDNA fragments. The cDNA undergone paired-end repair to convert overhangs into blunt ends. After 3'-monoadenylation and adaptor ligation, cDNAs were purified. Next, cDNA was amplified by PCR using primers specific for the ligated adaptors. (Illumina, Eindhoven, Netherlands). The generated libraries were submitted to quality control before being sequenced on an Illumina HiSeq 2500. RNA-seq data processing and analysis is described in Supplementary Methods.

**Proteomics processing and analysis**. EndoC-βH1 cells exposed or not to IFNα were extracted using the Metabolite, Protein and Lipid Extraction (MPLEx) approach. A detailed description of the method used for proteomics processing and analysis is provided in Supplementary Methods.

**Rank–rank hypergeometric overlap (RRHO) analysis**. To compare the signature induced by IFNα with the one present during insulitis in T1D individuals, we performed the RRHO mapping[15]. For this goal, a full list of log₂FC ranked genes from our RNA-seq of EndoC-βH1 cells and human islets (IFNα vs Control, 24 h) were compared against similarly ranked lists of purified primary beta cells obtained from individuals with T1D[16] and T2D[17] (T1D/T2D vs non-diabetic).

In a RRHO map, the hypergeometric $p$-value for enrichment of $k$ overlapping genes is calculated for all possible threshold pairs for each experiment, generating a matrix where the indices are the current rank in each experiment. The log-transformed hypergeometric $p$-values are then plotted in a heatmap indicating the degree of statistically significant overlap between the two ranked lists in that position of the map. Multiple correction was applied using the Benjamini–Yekutieli FDR correction.

**Dynamic regulatory events miner (DREM) modeling**. For reconstructing dynamic regulatory networks, we have used the DREM method[23], which integrates times series and static data using an Input-Output Hidden Markov Model (IOHMM), where the TF-DNA interaction information obtained from ChIP-seq experiments[48] was used as the input and our RNA-seq time series expression data as the output. A detailed description of DREM-based modeling is provided in Supplementary Methods.

**Weighted gene co-expression network analysis (WGCNA)**. On each dataset (RNA-seq and proteomics), we obtained modules of genes/proteins of similar expression profiles using WGCNA[28]. The soft threshold parameter for the RNA-seq dataset was chosen to be 10 (value to approximate a scale-free topology). Similar parameters were used for the analysis of RNA-seq of pancreatic human islets exposed or not to IFNα. Regarding the proteomics dataset, in order to achieve an approximated scale-free topology, we first normalized each protein expression in each temporal group (subtraction by mean and division by standard deviation), and then selected the soft threshold parameter as 14. After merging the modules

using a dissimilarity threshold of 0.25, we identified 8 modules in the RNA-seq dataset and 7 modules in the proteomics dataset.

To analyze module quality, we have used a set of statistics (density and separability metrics) from the *modulePreservation* function of the R package WGCNA[29]. For this purpose, we resampled the dataset 1000-times to create reference and test sets from the original data and evaluate module preservation, represented as the $Z_{summary}$ for each module across the resulting networks. $Z_{summary}$ > 2 indicates moderate preservation and $Z > 10$ high quality/preservation for each module[29]. To evaluate WGCNA module preservation in independent samples, we used the same R function, but in this case applying metrics based on module density and intramodular connectivity to give a composite statistic $Z_{summary}$.

To evaluate the overlap of RNA-seq and proteomics modules, we considered a mRNA to be differentially expressed at 8 or 24 h if its absolute fold-change was >1.5 and its FDR < 0.05. Regarding the proteomics dataset, we considered a protein to be differentially abundant at 8 or 24 h if the *t*-test *p*-value was <0.05. We selected only the differentially expressed genes/abundant proteins in the identified WGCNA modules. We then searched for the overlap between the elements of the RNA-seq and proteomics modules and obtained an overlap *p*-value (hypergeometric probability). We retained overlapping modules with a FDR < 0.05 and a minimum of 10 common elements.

**Protein–protein interaction network analysis**. The inBio Map protein–protein interaction (PPI) network database[31] was obtained from https://www.intomics.com/inbio/. We first restricted the network to contain only the elements expressed in human beta cells based on our RNA-seq database (mean RPKM > 0.5 in at least one condition). For each WGCNA overlapping module, we identified the proteins in the PPI network with a significantly high number of protein-to-protein connections to the set of elements in the module (FDR < 0.01, and minimum number of connections equal to 5). We considered only networks obtained for the overlapping modules #1, #2 and #5, as the other overlapping modules returned empty PPI networks. We then obtained PPI networks for each WGCNA overlapping modules, involving the original set of module elements, plus the respective identified connecting proteins. Communities of interacting proteins were identified using the EAGLE algorithm[76] with the following parameters: CliqueSize threshold: 6 and ComplexSize threshold: 2. Wordclouds of each community were generated using information from geneRIFs terms.

**Gene regulatory network analysis**. A network of regulatory interactions was obtained from RegNetwoks[30] (www.regnetworkweb.org). As in the PPI network, we first restricted the network to contain only the elements we found to be expressed in the RNA-seq dataset. Similarly to the PPI network analysis, for each WGCNA overlapping modules, we identified regulators with a significantly high number of regulatory connections to the set of elements in the module (FDR < 0.01, and minimum number of connections equal to 4). We then obtained regulatory networks for each WGCNA overlapping modules, involving the original set of module elements, plus the respective identified regulators.

To create a non-redundant dataset of motifs from regions of gained open chromatin, we used the compareMotifs.pl script from the package HOMER[77] to merge motifs with a similarity score threshold of 0.7. The remaining motifs were mapped to the gain open chromatin regions using the annotatePeaks.pl script.

**Transcription factor motif analysis**. Sequence composition analysis of de novo motifs was performed using findMotifGenome.pl from the package HOMER[77] with parameters '-size given -bits -mask'. The motifs having a $p \leq 10^{-12}$ and observed in >3.5% of the targets were chosen for subsequent analysis. All de novo matches having a similarity score to known TF motifs higher than 0.7 are shown in the tables (Fig. 4e and Supplementary Fig. 3c), or when no match was present over this threshold, the first hit was elected and its score is presented.

**Alternative splicing changes validation**. Alternative splicing changes identified from RNA-seq were validated by RT-PCR using specifically designed primers (Supplementary Table 3). To confirm cassette exons, the primers were adjacent to the predicted splicing event. This approach allowed us to discriminate between variants based on their fragment sizes. For alternative first exon usage (AFE) validation, we have designed primers spanning regions that are unique to the isoform of interest (Fig. 6g), and then normalized the results by the housekeeping gene β-actin. cDNA was amplified using MyTaq Red DNA polymerase (Bioline, London, UK), and PCR products were analyzed using an Agilent 2100 Bioanalyzer system (Agilent Technologies, Wokingham, U.K.). The molarity of each PCR band corresponding to a specific splice variant was quantified using the 2100 Expert Software (Agilent Technologies, Diegem, Belgium), and used to calculate the ratio inclusion/exclusion (SE) or isoform-X/β-*actin* (AFE).

**Small-RNA interference**. Transfection was performed using Lipofectamine RNAiMAX (Invitrogen) as described in Supplementary Methods. After that, the cells were kept in culture for a 48 h recovery period and subsequently exposed or not to IFNα as indicated. Supplementary Table 3 describes the sequences of siRNAs used in the present study.

**Real-time PCR analysis**. After harvesting of the cells, Poly(A) + mRNA was obtained using the Dynabeads mRNA DIRECT kit (Invitrogen) and reverse transcribed. Detailed description is provided in Supplementary Methods.

**Western blot, immunocytochemistry and flow cytometry**. Detailed description together with additional information on western blot, immunocytochemistry and flow cytometry analysis is provided in Supplementary Methods.

**Immunofluorescence**. After dewaxing and rehydration, samples were subjected to heat-induced epitope retrieval (HIER) in 10 mM citrate buffer pH 6.0, then probed in a sequential manner with appropriate antibodies as indicated in Supplementary Table 4. The relevant antigen–antibody complexes were detected using secondary antibodies conjugated with fluorescent dyes (Invitrogen, Paisley, U.K). Cell nuclei were stained with DAPI. After mounting, images were captured with a Leica AF6000 microscope (Leica, Milton Keynes, UK) and processed using the standard LASX Leica software platform (Version 1.9.013747). For quantification studies, randomly selected insulin-containing islets (ICIs) from individuals with or without diabetes were imaged, in addition to insulin-deficient islets (IDIs) from individuals with diabetes. Thirty ICIs were analyzed from 6 independent individuals (5 islets per individual), 20 IDIs were analyzed from 4 independent individuals (5 islets per individual) and 30 ICIs were analyzed from 6 independent control individuals (5 islets per individual). The mean fluorescence intensity (MFI) arising from detection of HLA-E was measured using LASX Leica quantification software.

**Therapeutic targets identification**. The top 150 upregulated genes shared among the RNA-seq of EndoC-βH1 cells and human islets exposed to IFNα for 24 h and the RNA-seq of beta cells[16] from T1D individuals were identified by the RRHO analysis. This list of genes was used to query the Connectivity Map dataset of L1000 cellular signatures, which has transcriptional responses of human cells to different chemical and genetic perturbations, using the CLUE platform (https://clue.io)[49]. To identify compounds potentially reverting the effects induced by interferons in beta cells, we have focused on perturbagens promoting signatures that were opposite (negative tau score) to our query list. Only perturbagens having a median tau score < −90 were considered for further evaluation.

Additionally, aiming at potential repurposing of drugs under clinical investigation for treatment of other pathologies, we have integrated the DrugBank database v5.1[51] with the PPI network obtained from WGCNA module #2 using the CyTargetLinker v4.0.0[78] within Cytoscape v3.6 to build a biological network annotated with drugs.

The small molecules and drugs pointed out by these two approaches were then validated in vitro as described above to verify their impact on IFNα-induced upregulation of cytokines/chemokines, ER stress markers, HLA class I and beta cell apoptosis.

**Cell viability assessment**. The cell viability is described in details in Supplementary Methods.

**Statistical analysis**. Data of the confirmatory experiments are expressed as means ± SEM. A significant difference between experimental conditions was assessed by paired *t*-test, unpaired *t*-test, one-way or two-ways ANOVA followed by Bonferroni correction for multiple comparisons as indicated using the GraphPad Prism program version 6.0 (www.graphpad.com). Results with $p \leq 0.05$ were considered statistically significant.

**Reporting summary**. Further information on research design is available in the Nature Research Reporting Summary linked to this article.

## Data availability

All raw and processed ATAC and RNA-sequencing data that support the findings of this study have been deposited in NCBI Gene Expression Omnibus (GEO) with the primary accession code GSE133221 (subseries are GSE133218: RNA-seq of EndoC-βH1 cells, GSE148058: RNA-seq of human islets, GSE133219: ATAC-seq of EndoC-βH1 cells). The proteomics datasets have been submitted to Pride under identifier number PXD014244 (http://www.ebi.ac.uk/pride/archive/projects/PXD014244). The network of regulatory interactions can be obtained from RegNetworks (http://www.regnetworkweb.org/download/RegulatoryDirections.zip). The DrugBank database v5.1 can be downloaded from: https://www.drugbank.ca/releases/5-1-0/downloads/all-full-database. The inBio Map protein–protein interaction (PPI) network database can be obtained from: https://www.intomics.com/inbio/api/data/map_public/2016_09_12/inBio_Map_core_2016_09_12.zip. The CAGE peaks from FANTOM5 database can be obtained on: http://fantom.gsc.riken.jp/5/datafiles/phase2.5/extra/CAGE_peaks/. The Connectivity Map database can be accessed using the CLUE platform (https://clue.io). The RNA polymerase II (POLR2A) ChIP-seq of human K562 cells can be obtained from the ENCODE project (GSM935474, https://www.encodeproject.org/experiments/ENCSR000FAX/). The Exon Ontology database can be accessed from: http://fasterdb.ens-lyon.fr/ExonOntology/. The information about T1D risk genes can be found on immunobase (www.immunobase.org) and GWAS catalog (https://www.ebi.ac.uk/gwas/).

The source data underlying Figs. 2c–m, 3a–e, g, i, 5d, g, j, 6b, d, 7c–f, 8b–e, 9a, b and Supplementary Figs. 4e–h, 5c–m, 6b, d, 12a–c, 12e–g, 13c–d are provided as a Source data file.

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

## Acknowledgements
We are grateful to Dr. Andrea A. Schiavo for help in the implementation of the protocol for FACS-purified human beta cells isolation and Isabelle Millard, Anyishaï Musuaya, Nathalie Pachera and Michaël Pangerl of the ULB Center for Diabetes Research for excellent technical support. This work was supported by grants from the Fonds National de la Recherche Scientifique (FNRS), Welbio CR-2015A-06 and CR-2019C-04, Belgium; the Horizon 2020 Program, T2Dsystems (GA667191); the National Institutes of Health, NIH-NIDDK-HIRN Consortium 1UC4DK104166-01 and Innovate2CureType1 - Dutch Diabetes Research Fundation (DDRF). D.L.E. and P.M. have received funding from the Innovative Medicines Initiative 2 Joint Undertaking under Grant Agreement No. 115797 (INNODIA). This Joint Undertaking receives support from the Union's Horizon 2020 research and innovation programme and "EFPIA", "JDRF" and "The Leona M. and Harry B. Helmsley Charitable Trust". The DiViD study is funded by The South-Eastern Norway Regional Health Authority (grant to K.D.-J.), the Novo Nordisk Foundation (grant to K.D.-J.), and through the PEVNET (Persistent Virus Infection in Diabetes Network) Study Group funded by the European Union's Seventh Framework Programme (FP7/2007-2013) under grant Agreement Number 261441 PEVNET. Additional support was from a JDRF Career Development Award (5-CDA-2014-221-A-N) to S.J.R., a JDRF research grant awarded to the network of Pancreatic Organ Donors – Virus (nPOD-V) consortium (JDRF 25-2012-516); an MRC Project Grant MR/P010695/1 awarded to S.J.R. and N.G.M., a studentship grant from the Norman Family Trust (to S.J.R. and N.G.M.), a Spanish Ministry of Economy and Competiveness (SAF2017-86242-R to L.P.) and Marató TV3 (201624.10 to L.P.). L.P. is a recipient of a Ramon y Cajal contract from the Spanish Ministry of Economy and Competitiveness (RYC-2013-12864) and M.R. is supported by an FI Agència de Gestió d'Ajuts Universitaris i de Recerca (AGAUR) PhD fellowship. Part of the work was performed in the Environmental Molecular Sciences Laboratory, a U.S. Department of Energy (DOE) national scientific user facility at Pacific Northwest National Laboratory (PNNL) in Richland, WA. Battelle operates PNNL for the DOE under contract DE-AC05-76RLO01830.

## Author contributions
M.L.C., L.P., T.O.M. and D.L.E. conceived, designed and supervised the experiments. M.L.C., M.I.A., J.L.E.H., A.C.B., A.C., H.R.V, E.S.N., N.G.M. and S.J.R. performed and analyzed the experiments. L.K., K.D.-J., M.L., M.A.R, J.J.-M., L.M. and P.M. contributed with material and reagents. M.L.C., M.R.R., B.-J.M.W.-R. and J.-V.T performed bioinformatic analyses. M.L.C. and D.L.E. wrote the manuscript, and all authors revised it. M.L.C. and D.L.E. are the guarantors of this work and, as such, have full access to all the data in the study and take responsibility for the integrity of the data and the accuracy of the data analysis.

## Competing interests
The authors declare no competing interests.
