## [Peer Review File · Nature Communications]

Reviewers' comments:

Reviewer #1 (Remarks to the Author):

In this paper Colli et al have exposed human beta cells to IFN- α and studied chromatin organization, gene expression and protein expression. Parts of the paper are interesting but there are also limitations.

1. It is not fully clear why they only study IFN- α and not other cytokines in this paper. It seems logical to also include other cytokines. Moreover, the impact of other cytokines on similar outcomes are studied by the same group of researchers in a paper bioRxiv and it is not clear why they split these papers into two.
2. It is unclear what the authors have done and they want to describe with the first section of the results and with the legend to Sup Figure 1a. Please clarify and present the data in Sup Figure 1 in a more detailed and convincing manner.
3. The title is misleading. They have not studied human pancreatic beta cells but rather a fetal cell line and human islets.
4. Merging data suggest limitations with the EndoC- β H1 β cells since they are not fully mature beta cells and it would be better with mature human beta cells from for example sorted beta-cells from human islets. Dispersed islets include too many different cell types.
5. I could not find a table with the 4,400 and 1000 gained OCRs: Please include and refer to on page 6. Also include in the table the association between OCRs and expression of individual genes.
6. Please include number of samples in the legend to each figure. Currently it is difficult to follow how many samples were used at each figure (for a few figures it is clear).
7. For Sup figure 3c,d did the authors use raw p-value of <0.05 ? Did they correct for multiple testing?
8. For knockdown in sup fig 4, did the authors validate the knockdowns with western blot? If not, then please include western blot.
9. Has the data in figure 2m been published previously? If it has, do not present published data in figures or sup figures, only refer to data.
10. The analysis of splicing is interesting. They also try to find RNA binding protein that may regulate these events. It would have been interesting to silence some of them to test if that affected the splice events.
11. Did disease duration and or age of the patient in sup table 5 affect the results in the paper?
12. The number of samples in each genome-wide analysis is very small e.g. $n=4$ and 5. It would therefore require further validation in independent set of samples/cohorts.
13. QC data for the ATAC-seq data is required. Also present the number of and specific peaks identified for each sample and tape station data for each sample.
14. They exposed cells and islets to 50 U/ml IFN- α based previous dose response studies. Is this a concentration that is also seen in people in vivo and therefore clinically relevant?
15. They should test one more housekeeping gene for qPCR since beta-actin is not always stable.

Minor:

1. Page 4, replace "where" with "were"?

Reviewer #2 (Remarks to the Author):

Summary comments: the authors present a compelling case for using EndoC- β H1 cells exposed to INF α as a model for T1D β -cells based upon strong overlap in differential gene and protein expression between the treated cell line and the actual human samples. In so doing, they identify transcription factors and chromatin responses that impact immune response genes and control their expression in response to cytokine exposure. By identifying modules of genes/proteins that are upregulated, they were able to test small molecules that could antagonize cytokine-specific response and reduce MHC Class I expression and apoptosis, which helps translate bioinformatic information to biological and potential therapeutic application. Overall, the manuscript is well organized and of value to the pancreatic islet field and a larger audience because of its applied methodology and biological findings. Despite this positive impression, the comments below generally request more biological detail so that the interesting scientific findings are not lost among complex and not very descriptive bioinformatic figures.

Specific Comments on Results (these are generally organized by figure)

Supplementary Figure 1: the text of the paper mentions significant intersection following RRHO analysis of ranked lists (p.4). Could the authors elaborate on the number and percentage of genes that significantly overlapped between their data (EndoC- β H1 + INF α) versus the data from the cited sources of T1D beta cells, T1D islets, and T2D beta cells and provide the lists of genes they tested for overlap (in addition to the RRHO maps presented in Figures 1B-D). This is helpful to further evaluate the effectiveness of this cell-line and treatment as a model for T1D beta cells in addition to their control comparison versus T2D beta cells. Since this comparison is the whole basis for the rest of the paper, a clearer evaluation/comparison of the model and specific samples would more strongly support the findings.

General comment: the authors interchange between EndoC- β H1 cells and pancreatic beta cells, especially at the beginning of the Results section when describing figures 1 and 2. This is a bit confusing since most of the experiments are completed with the cell line, but some experiments with human pancreatic islets. It would be helpful to be clearer about this topic. (In other words, use the term cell line when referring to experiments with the EndoC- β H1 cells and reserve human beta cells for examination of beta cells from human islets.) To provide two examples: Figure 1A shows "Human beta cells" in the schematic, but the whole experiment was done on EndoC- β H1 cells, which is mentioned in the figure. Similarly, Figure 2G/I was done with EndoC- β H1 cells and 2H/J was done with islets (this is labeled and described clearly), however the next manuscript text about figures 2K/L mentions human beta cells, but the figure legend highlights that they're from the EndoC- β H1 cell line.

Figure 1: Figures C-E refer to a proportion of transcripts that are up/down regulated or demonstrate equal/no expression. How many genes are examined in these instances? The DE genes are listed in the supplementary tables, but it's not clear the number of genes that related to the corresponding ATAC-Seq data.

Figure 2: In figure 2N, representative images of specific islets are shown. In 2O, could the authors elaborate and show how many donors contributed how many islets to the ND, T1D ICI, and T1D IDI samples? Furthermore, were an equal number (or percentage) chosen from each islet sample shown in Supplementary Table 5?

Figure 3: In the module development figures derived from the WGCNA package that show how many of the transcripts and proteins overlap in their expression, what time period (2h, 8h, 24h or a combination) was used to make this analysis – in the splicing analysis, 8h and 24h were used. What % of genes/proteins that are used in the module overlap with the original DEG or DAP lists? The overall correlation in Figure 3C is striking (i.e. they all are either both up or down regulated), so knowing how the genes and proteins were selected/removed by prior to running the WGCNA modules would be helpful.

Figure 4: this is a similar question to the above – of the 343, 1690, and 1663 genes that were identified, how many of them were unique to their specific 2 hour, 8 hour, and 24 hour time period versus how many of them overlapped? Since Figure 1 spends some time showing the proportion of genes up or down regulated in response to open chromatin regions and a correlation was made between upregulated genes and proteins, it would be informative to know how many genes/chromatin regions stay open throughout the exposure to INF α versus how many are quick or delayed response.

Figure 6: These experiments are all quite striking in their results – there's both a marked decrease in the gene expression of surface proteins and transcription factors following exposure to the identified family of small molecules. A further application and demonstration that the model system effectively represented what happens in human islets would greatly strengthen the paper. Would it be possible to quantitate the reduction in MHC Class I expression in the islets (6K) and perform the same experiments with human islets to examine CXCL10, CHOP, and apoptosis similar to the way it was done for EndoC- β H1 cells?

Thank you for the opportunity to review your exciting research. David Blodgett

Reviewer #3 (Remarks to the Author):

The manuscript is an impressive amount of work, with multiple data sets and numerous multi-omics analyses, to characterize the human beta cell response to treatment to IFN α , which is expressed in the islets of T1D individuals, and shown to be regulated by genetic and environmental factors associated with T1D.

General

1. The manuscript is very dense with a lot of information but there seems to be some missing motivation (see below under Study Design), and overview information (e.g., see below under analysis regarding gene-protein overlap).
2. It is not clear what the novelty of this contribution. Is it the type of cells? the types of analysis? The authors should motivate how this work is unique.

Study Design

3. Why are all data sets not collected at all 3 time points (e.g., proteomics missing at 2h, ATACseq missing at 8 h).
4. Why are the 2h, 8h and 24h time points the most relevant, how were they selected?

Analysis

5. A lot of the analyses is based on the RNA-Seq data. The authors use RPKM as normalized sequence count values. However, RPKM has been recognized to have major weaknesses, even by the authors of RPKM, it is now more common to use methods such as TPM (or TMM). Do your conclusions change if one of these alternatives are used instead?
6. What is the overlap between the genes/protein products that are measured using RNA-Seq and proteomics, and overlap with the transcripts/proteins that are significantly changing?
7. It is not clear if WGCNA was run with all time points & treated/untreated samples. If so, the sample size appears to be 20 for the RNA-Seq ($n=5 \times 2 \text{ time points} \times 2 \text{ treatments} = 20$) and 16 for the protein ($n=4 \times 2 \text{ time points} \times 2 \text{ treatments}$). This is on the low end to run a method like WGCNA, and there are concerns about the stability of the results. Did the authors do any sensitivity analysis?

8. Also for WGCNA, if multiple time points by treatments are used to define co-expression, but what if co-expression is broken due to treatment? Then the resulting WGCNA modules would only be capturing when co-expression is constant across time points and treatments.

Point by point Answers to the Reviewers' comments:

Reviewer #1 (Remarks to the Author):

In this paper Colli et al have exposed human beta cells to IFN- α and studied chromatin organization, gene expression and protein expression. Parts of the paper are interesting but there are also limitations.

We thank the Reviewer for considering our paper interesting, and for pointing relevant limitations that we have addressed in the revised version of the manuscript.

1. It is not fully clear why they only study IFN- α and not other cytokines in this paper. It seems logical to also include other cytokines. Moreover, the impact of other cytokines on similar outcomes are studied by the same group of researchers in a paper bioRxiv and it is not clear why they split these papers into two.

The cytokines IL1 β + IFN γ , studied in a previous multi-omics study by our groups ¹, and IFN α contribute to the pathogenesis of type 1 diabetes at different stages of the disease, i.e. while IFN α is crucial in the early and innate immune phase of T1D, IL1 β + IFN γ intervene at the later and adaptive phase of the autoimmune response ^{2,3}. Based on this, and on the fact that the impact of these cytokines is different in human beta cells, in our previous ^{1,4,5} and present work we studied these cytokines separately.

Of note, the present paper, besides studying a different cytokine as compared to Ramos-Rodriguez M *et al.* ¹, covers some different aspects, namely alternative splicing and drug discovery. Last, but not least, we did not “split these papers in two”, but developed them independently as two separate studies, each requiring 4-5 years of work and generating a very large amount of data (e.g. 19 figures plus 12 large tables in the present paper and a similar amount of data at Ramos-Rodriguez M *et al.* ¹).

2. It is unclear what the authors have done and they want to describe with the first section of the results and with the legend to Sup Figure 1a. Please clarify and present the data in Sup Figure 1 in a more detailed and convincing manner.

The goal of the first section of Results and the legend of the now **Supplementary Fig. S2**, is to show the potential relevance of our findings to the clinical situation. For this purpose, we took two approaches: 1. Examine whether candidate genes for T1D expressed in human islets regulate IFN signaling (**Supplementary Fig. S2A**); 2. Compare our *in vitro* data of IFN α -treated EndoC- β H1 cells (and now also human islets; see new **Supplementary Fig. S1**) with available RNA-seq data of human beta cells from T1D patients, to check whether the global changes in gene expression observed are similar to the human disease situation (**Supplementary Figs. S2F and G**). This information is important to show the potential relevance of our findings to the human disease. We have now better clarified the text of Results (page 04, paragraph 01) and the legend for **Supplementary Fig. S2**.

3. The title is misleading. They have not studied human pancreatic beta cells but rather a fetal cell line and human islets.

To address this relevant concern we have now added RNA sequencing studies from 6 independent human islet preparations containing in average > 50% of beta cells ($53 \pm 7\%$ (mean \pm SD), evaluated by immunocytochemistry as described in Methods) exposed to IFN α at similar time points (new **Supplementary Fig. S1**). Furthermore, we also validated the impact of IFN α on primary human beta cells by using a recently described method to purify live human beta cells ⁶ (see Methods) allowing us to obtain preparations

with > 80 % primary beta cells (**Supplementary Figs. S6A and B**). This, and all the other experiments performed with human islets (see **Figures 2, 3, 4 and 5 and Supplementary Fig. S13**), support the use of “human pancreatic beta cells” in the title.

4. Merging data suggest limitations with the EndoC-βH1 β cells since they are not fully mature beta cells and it would be better with mature human beta cells from for example sorted beta-cells from human islets. Dispersed islets include too many different cell types.

We and others groups ⁷⁻⁹ have shown that the EndoC-βH1 responses to different stimuli are mostly similar to the ones observed in primary human islets ^{1,4,5,10,11}, particularly regarding responses to pro-inflammatory cytokines ^{1,4,5,12}.

To address the Reviewer’s concern, we have now performed new and targeted experiments where FACS-sorted human beta cells were exposed to IFNα (see new **Figure 2I and Supplementary Figs. S6A to C**; of note, sorting adult human beta cells is a not trivial procedure, and we only managed to have the method running in our lab by late 2019). Exposure of FACS-purified human beta cells to IFNα generated very similar results to EndoC-βH1 cells confirming the upregulation of genes related to antigen presentation (*HLA-I*), antiviral responses (*MX1*, *MDA5*), ER stress (*CHOP*), immune cells recruitment (*CXCL10*) and checkpoint regulators (*HLA-E* and *PDL1*) (**Figure 2I and Supplementary Fig. S6C**).

5. I could not find a table with the 4,400 and 1000 gained OCRs: Please include and refer to on page 6. Also include in the table the association between OCRs and expression of individual genes.

We have now attached **Supplementary Table S2**, in which we list the OCRs at 2 and 24 hours, their fold changes, their classification and the annotation of the TSS of protein-coding genes located < 20Kb from the OCR.

6. Please include number of samples in the legend to each figure. Currently it is difficult to follow how many samples were used at each figure (for a few figures it is clear).

We have now indicated the number of samples in the legends for all figures.

7. For Sup figure 3c,d did the authors use raw p-value of <0.05? Did they correct for multiple testing?

We have now adjusted the **Figures 3C and D** and its legend by presenting genes/proteins with a False Discovery Rate (FDR) < 0.05.

8. For knockdown in sup fig 4, did the authors validate the knockdowns with western blot? If not, then please include western blot.

These siRNAs have been validated in previous studies by our group ^{4,12}. We have now included the confirmation of the knockdown efficiency at the protein level for IRF1, STAT1 and STAT2 by performing Western blot and densitometry analysis. The results are presented in **Supplementary Fig. S5** for the different conditions studied.

9. Has the data in figure 2m been published previously? If it has, do not present published data in figures or sup figures, only refer to data.

These datasets have been previously published at ¹³, but without any specific mention to HLA-E. We have accessed the raw data, and re-calculated the expression HLA-E for the present study. Following the Reviewer's advice, we have now deleted **Figure 2M**, and mention the information in the text with the supporting reference.

10. The analysis of splicing is interesting. They also try to find RNA binding protein that may regulate these events. It would have been interesting to silence some of them to test if that affected the splice events.

In order to address the Reviewer's suggestion, we have first reproduced the downregulation promoted by IFN α in two RNA-binding proteins (RBPs), namely ELAVL1 and HNRNPA1, using siRNAs targeting specifically these genes (**Supplementary Figs. S12A and E**) in a similar approach as we have done for another RBP, SRp55 ¹⁴. Next, we evaluated whether this inhibition reproduced the changes induced by IFN α in the exon usage of FHL1 and CAPRIN2 (**Supplementary Figs. S12B and F**) two known targets of ELAVL1¹⁵ and HNRNPA1¹⁶, respectively. In line with this, silencing of these RBPs produced effects on exon usage (**Supplementary Figs. S12C and G**) that were very similar to IFN α treatment (**Supplementary Figs. S12B and F**). These new data are now described in Results, page 17 of the manuscript.

11. Did disease duration and or age of the patient in sup table 5 affect the results in the paper?

There was no impact of age on HLA-E expression among the T1D donors or controls. Regarding disease duration, all diabetic donors had a short duration of disease (the longest being 18 months post diagnosis) (see **Supplementary Table S9**) and there were no obvious differences on HLA-E expression during this window of time.

12. The number of samples in each genome-wide analysis is very small e.g. n=4 and 5. It would therefore require further validation in independent set of samples/cohorts.

To address this concern, we have now added data on RNA-sequencing of six independent human islet preparations (see new **Supplementary Figs. S1, S2 and S8**).

Of note, in this study we tested 4 independent biological replicates per condition for a total of 16 ATAC-seq experiments. In order to test if the number of samples used was sufficient to cover inter-replicate variation, we computed saturation curves for the ATAC-seq data (see below **Figure R1**). For both time points this analysis shows that by using 4 replicates we could approach saturation of the number of peaks detected, indicating that our experiments are capturing most of the inter-replicate variation.

Figure R1 (for the Reviewers only). Variability of the ATAC-seq experiments. Each plot shows the merged of the peaks called when combining different peak sets obtained from independent ATAC-seq experiments (treated and untreated at each time point). Each independent combination is represented as a colored dot, the black dot represents the mean of all combinations and the solid black line the standard deviation of all the combinations.

These analyses supports the solidity of the present results. We nevertheless repeated the ATAC-seq analyses in independent datasets of human pancreatic samples and compared them with the results obtained in the present study in EndoC- β H1 cells (see attached **Figure R2**). The results confirm a very high overlap between ATAC-seq peak sets obtained from different beta cell lines and human islet samples with those obtained in the present study.

Figure R2 (for the Reviewers only). Overlap between ATAC-seq peaks identified from published datasets ^{1,17-19} with the peak set generated in the present study. Peaks were called for the public datasets using MACS2 ²⁰ with Q-value < 0.0001 and fold-change > 4.

Of note, we have used a similar number of samples (n = 4 - 5) for the genome-wide approaches in our recent publication in Nature Genetics ¹, based on a similar approach for validation of the results.

13. QC data for the ATAC-seq data is required. Also present the number of and specific peaks identified for each sample and tape station data for each sample.

Please find below in **Figure R3** showing the ATAC-seq quality control measures taken to validate our approach. A heatmap in **Figure R3a** illustrates how the samples were deeply sequenced (mean library size is 168M reads), the alignment rates were adequate (mean alignment rate is 82%), the proportion of mitochondrial reads was similar to ATAC-seq assays from other works ¹ (mean percentage of mitochondrial reads 35%) and the normalized strand cross-correlation coefficient (NSC), and relative strand cross-correlation coefficient (RSC), values reflect the high quality of the experiments.

As an additional quality measures, in **Figure R3b** we show that the genome-wide correlation between replicates in different conditions and time points was very high. Moreover, in **Figure R3c** we represent the expected enrichment of ATAC-seq reads around the promoters of coding genes, compared to a randomized set of regions and in

Figure R3d we show the signal-to-noise ratios, representing the number of reads that are found inside called peaks compared to the total number of peaks.

All the above-mentioned measures support the high quality of the present ATAC-seq experiments.

Figure R3 (for the Reviewers only). ATAC-seq quality control. **a.** Heatmap showing the values for different ATAC-seq quality measures: total library size, percentage of alignment, percentage of mitochondrial reads, normalized strand cross-correlation coefficient (NSC, values significantly lower than 1.1 (<1.05) tend to have low signal to noise or few peaks) and relative strand cross-correlation coefficient (RSC, values significantly lower than 1 (<0.8) tend to have low signal to noise). **b.** Person's correlation between replicates in different conditions and time points. **c.** Mean ATAC-seq reads around the transcription start site (TSS) of protein-coding genes compared to randomized sites. **d.** Mean signal-to-noise ratios representing the percentage of the total reads located inside called peaks compared to a random set of peaks. The bar represents the standard deviation.

As requested, we are also attaching the TapeStation results for each ATAC-seq samples in **Figure R4**, showing the nucleosomal pattern in ATAC-seq samples.

Figure R4 (for the Reviewers only). TapeStation results for ATAC-seq samples. This plot shows that the distribution of ATAC-seq fragments resembles the nucleosomal pattern, with the first peak at ~150bp corresponding to single nucleosomes and the subsequent ones to dimers and trimers.

As requested by the Reviewer, we added in **Supplementary Table S2A** the exact number of peaks called for each ATAC-seq sample, together with the actual coordinates of these peaks (narrowPeak files) (see below).

Sample	# Peaks
ctrl-24h 1	130,165
ctrl-24h 3	106,221
ctrl-24h 4	104,803
ctrl-24h 5	151,509
ctrl-2h 1	141,947
ctrl-2h 3	113,758
ctrl-2h 4	132,518
ctrl-2h 5	135,921
ifna-24h 1	134,327
ifna-24h 3	94,704
ifna-24h 4	200,155

ifna-24h 5	153,899
ifna-2h 1	138,451
ifna-2h 3	129,876
ifna-2h 4	125,274
ifna-2h 5	152,364

Information added to Supplementary Table S2A. Number of peaks called for each sample. Peaks are called from ATAC-seq data using MACS2²⁰.

14. They exposed cells and islets to 50 U/ml IFN- α based previous dose response studies. Is this a concentration that is also seen in people in vivo and therefore clinically relevant?

We have used the dose of 2,000U/ml (equivalent to 11.1 pg/ml) of IFN α and 50U/ml (equivalent to 240 pg/ml) of IL1 β based on our previous dose-response experiments performed on EndoC- β H1 cells⁴ (see Figure R5). A previous study demonstrated that the serum levels of IFN α are significantly higher in T1D individuals in comparison to healthy individuals with a mean of ~ 5 pg/ml (range: 0-30 pg/ml)²¹. The authors also attempted to estimate the local IFN α concentrations present during a diabetogenic viral infection of the pancreas. For this purpose they measured the concentrations of IFN α produced by peripheral blood mononuclear cells (PBMCs) from T1D patients stimulated with Coxsackievirus B4, a strain previously identified in the pancreas of T1D patients²², obtaining mean values of ~ 30 pg/ml (range: 0-65 pg/ml). Similar values (range 0-53 pg/ml) were observed locally during central nervous system viral infections²³. Regarding IL1 β , patients with severe and complicated hepatitis C have serum IL1 β concentrations in the range of 0.7 – 187 pg/ml²⁴. Beta cell loss during the pathogenesis of T1D occurs in the context of a localized immune response, where cytokines are released on a cell-to-cell basis in the so-called “immune synapse”. The target cells are thus exposed to higher concentrations of cytokines than those observed in the general circulation. Considering all this information, the concentration of IFN α used in our present study is in the range of values observed during potentially diabetogenic viral infections or in viral infections in the central nervous system. This information is now added to the Supplementary Methods on page 01.

[REDACTED]

15. They should test one more housekeeping gene for qPCR since beta-actin is not always stable.

The point is well taken. We have now tested a second housekeeping gene, Glyceraldehyde-3-Phosphate Dehydrogenase (*GAPDH*), which was not modified by $\text{IFN}\alpha$ in our RNA-seqs of both EndoC- β H1 cells and pancreatic human islets. The results obtained using *GAPDH* as the housekeeping gene in both EndoC- β H1 cells and human islets were very similar to the previous ones using *Beta-actin* as the housekeeping gene in EndoC- β H1 cells and human islets (**Figure R6 below**).

Figure R6 (for the Reviewers only). Comparison of housekeeping genes. Endo- β H1 cells (**A and B**) or human islets (**C and D**) were exposed or not to IFN α for 24h and mRNA expression of *HLA-I* (ABC), *HLA-E*, *CXCL10*, *PDL1*, and *CHOP* was evaluated by real-time RT-PCR. The values were normalized by two different housekeeping genes: *Beta-actin* (**A and C**) or *GAPDH* (**B and D**), and then by the highest value of each experiment considered as 1 ($n = 4 - 6$, ** $p < 0.01$, *** $p < 0.001$, paired t test).

We also evaluated the effect of the new housekeeping gene in experiments using different siRNAs and time points of IFN α exposure obtaining also similar results using *Beta-actin* (**Figure R7A**) or *GAPDH* (**Figure R7B**). These results support the choice of *Beta-actin* as the housekeeping gene for our IFN α model.

Figure R7 (for the Reviewers only). Comparison of housekeeping genes. A and B. Endo- β H1 cells were transfected with an inactive control siRNA (siCT) or siRNAs

targeting *IRF1* (siIRF1), *STAT1* (siSTAT1), *STAT2* (siSTAT2) or *STAT1* plus *STAT2* (siSTAT1+2). After 48h of recovery, the cells were exposed to IFN α for 2 or 24h and PDL1 mRNA expression was determined by real-time RT-PCR (**A**, **B**). The values were normalized by the housekeeping genes *Beta-actin* (**A**) or *GAPDH* (**B**) and then by the highest value of each experiment considered as 1 (n = 3 - 5, **p < 0.01, ***p < 0.001, ANOVA with Bonferroni correction).

Minor:

1. Page 4, replace “where” with “were”?

Corrected as indicated.

Reviewer #2 (Remarks to the Author):

Summary comments: the authors present a compelling case for using EndoC- β H1 cells exposed to INF α as a model for T1D β -cells based upon strong overlap in differential gene and protein expression between the treated cell line and the actual human samples. In so doing, they identify transcription factors and chromatin responses that impact immune response genes and control their expression in response to cytokine exposure. By identifying modules of genes/proteins that are upregulated, they were able to test small molecules that could antagonize cytokine-specific response and reduce MHC Class I expression and apoptosis, which helps translate bioinformatic information to biological and potential therapeutic application. Overall, the manuscript is well organized and of value to the pancreatic islet field and a larger audience because of its applied methodology and biological findings. Despite this positive impression, the comments below generally request more biological detail so that the interesting scientific findings are not lost among complex and not very descriptive bioinformatic figures.

We are grateful to the Reviewer for finding our manuscript of value, and for providing detailed and relevant suggestions for improvement.

Specific Comments on Results (these are generally organized by figure)

Supplementary Figure 1: the text of the paper mentions significant intersection following RRHO analysis of ranked lists (p.4). Could the authors elaborate on the number and percentage of genes that significantly overlapped between their data (EndoC- β H1 + INF α) versus the data from the cited sources of T1D beta cells, T1D islets, and T2D beta cells and provide the lists of genes they tested for overlap (in addition to the RRHO maps presented in Figures 1B-D). This is helpful to further evaluate the effectiveness of this cell-line and treatment as a model for T1D beta cells in addition to their control comparison versus T2D beta cells. Since this comparison is the whole basis for the rest of the paper, a clearer evaluation/comparison of the model and specific samples would more strongly support the findings.

We have now included a table presenting the complete list of genes used in each comparison for the RRHO analysis (**Supplementary Table S1**) and added the descriptive information about the number and percentage of overlapping genes in Results, page 4. We have also included RRHO comparisons using the new RNA-seq of human islets exposed to INF α , which support the initial findings obtained with EndoC- β H1 cells (**Supplementary Figs. S2F and G**).

General comment: the authors interchange between EndoC- β H1 cells and pancreatic

beta cells, especially at the beginning of the Results section when describing figures 1 and 2. This is a bit confusing since most of the experiments are completed with the cell line, but some experiments with human pancreatic islets. It would be helpful to be clearer about this topic. (In other words, use the term cell line when referring to experiments with the EndoC- β H1 cells and reserve human beta cells for examination of beta cells from human islets.)

To provide two examples: Figure 1A shows “Human beta cells” in the schematic, but the whole experiment was done on EndoC- β H1 cells, which is mentioned in the figure. Similarly, Figure 2G/I was done with EndoC- β H1 cells and 2H/J was done with islets (this is labeled and described clearly), however the next manuscript text about figures 2K/L mentions human beta cells, but the figure legend highlights that they’re from the EndoC- β H1 cell line.

We have followed the Reviewer’s suggestion and now make it clear which experiments were performed with EndoC- β H1 cells (named EndoC- β H1 cells or beta cell line), with human pancreatic islets or FACS-purified human beta cells (named human beta cells).

Figure 1: Figures C-E refer to a proportion of transcripts that are up/down regulated or demonstrate equal/no expression. How many genes are examined in these instances? The DE genes are listed in the supplementary tables, but it’s not clear the number of genes that related to the corresponding ATAC-Seq data.

For this analysis we selected a total of 19,430 protein-coding genes from Gencode v18²⁵. As specified in Methods section, we annotated ATAC-seq peaks to genes by associating an open chromatin region to a gene if its TSS is < 20kb from the center of such peak (see below, **Figure R8**).

Using this approach, we are able to annotate ~20% of all detected OCRs to a gene at each time point. We annotated a median of 3 OCRs to a gene (range 1-15), being 96% of the protein coding genes effectively annotated to at least one OCR (**Figure R8**).

Overall such results are in line with previous studies annotating open chromatin regions to protein coding genes¹.

Figure R8 (for the Reviewers only). Numbers of OCR and their associated genes. Bar plots showing the percentage of the OCRs and genes located at <20Kb from each other. The labels represent the specific numbers.

Figure 2: In figure 2N, representative images of specific islets are shown. In 2O, could the authors elaborate and show how many donors contributed how many islets to the ND, T1D ICI, and T1D IDI samples? Furthermore, were an equal number (or percentage) chosen from each islet sample shown in Supplementary Table 5?

To produce the data shown in **Figure 2O**, 30 ICIs were analysed from 6 independent individuals with T1D (5 islets per individual; middle group), 20 IDIs were analysed from 4 independent individuals with T1D (5 islets per individual; right hand group), and 30 ICIs were analysed from 6 independent individuals without diabetes (5 islets per individual; left hand group). The mean fluorescence intensity (MFI) arising from detection of HLA-E was measured using LASX Leica quantification software. This information is now included in the legend of **Figure 2**.

Figure 3: In the module development figures derived from the WGCNA package that show how many of the transcripts and proteins overlap in their expression, what time period (2h, 8h, 24h or a combination) was used to make this analysis – in the splicing analysis, 8h and 24h were used. What % of genes/proteins that are used in the module overlap with the original DEG or DAP lists? The overall correlation in Figure 3C is striking (i.e. they all are either both up or down regulated), so knowing how the genes and proteins were selected/removed by prior to running the WGCNA modules would be helpful.

To generate the WGCNA modules we have evaluated the samples from 8 and 24h since these are the time points where both RNA-seq and proteomics data are available. We now have added to **Supplementary Fig. S7C** a graph showing the number of non-modified, upregulated and downregulated genes present in each cluster identified in the RNA-seq and proteomics datasets. In the analysis of **Figure 3B** we have used 49% of all the

protein-coding genes differently expressed (DEG) induced at 8 or 24h, and 89% of the differentially abundant proteins (DAP) identified at the same time points. This information is now included in Results, page 13 of the manuscript.

Figure 4: this is a similar question to the above – of the 343, 1690, and 1663 genes that were identified, how many of them were unique to their specific 2 hour, 8 hour, and 24 hour time period versus how many of them overlapped? Since Figure 1 spends some time showing the proportion of genes up or down regulated in response to open chromatin regions and a correlation was made between upregulated genes and proteins, it would be informative to know how many genes/chromatin regions stay open throughout the exposure to $IFN\alpha$ versus how many are quick or delayed response.

Regarding OCRs dynamics we observe that most of the regions that gained chromatin accessibility at 24h were already gained at 2h (early response), while only 10% of all gained regions were specifically gained at 24h (late response). From these analyses we can thus conclude that most of the changes in chromatin accessibility happened early (< 2h) after the exposure to $IFN\alpha$.

On the other hand the profile of differentially expressed transcripts (DETs) induced by $IFN\alpha$ demonstrated that 4.1%, 32% and 31.7% of all the DET were exclusively induced by $IFN\alpha$ at 2h (early response), 8h (intermediary response) and 24h (late response), respectively (**Figure R9**). These findings demonstrate that the main effects of $IFN\alpha$ on the transcriptional activity are observed during intermediary or late time points. We thank the Reviewer for the comment, which allowed us now to clarify in Results, pages 6 and 16, these important aspects of chromatin and mRNA transcription dynamics.

Figure R9 (for the Reviewers only). Overlap of differentially expressed transcripts. Venn diagram showing the differentially expressed transcripts induced by $IFN\alpha$ in the RNA-seq of EndoC- β H1 cells. The percentages are proportional to the total number of DETs identified in in all the time points (FDR < 0.05 and $|\log_2FC| > 0.58$, n = 5 independent experiments).

Figure 6: These experiments are all quite striking in their results – there's both a marked decrease in the gene expression of surface proteins and transcription factors following exposure to the identified family of small molecules. A further application and demonstration that the model system effectively represented what happens in human islets would greatly strengthen the paper. Would it be possible to quantitate the reduction in MHC Class I expression in the islets (6K) and perform the same experiments with human islets to examine CXCL10, CHOP, and apoptosis similar to the way it was done for EndoC-βH1 cells?

We have now validated these key findings in human islets and observed a similar impact of the JAK inhibitor baricitinib on IFNα-induced mRNA expression of *HLA class I*, *CXCL10*, *CHOP* and cell apoptosis (**Figures 6I and N**). We also quantified MHC class I protein expression from the immunocytochemistry samples of human islets exposed to IFNα in the presence or not of baricitinib confirming the results observed at the mRNA level (**Figure 6K**).

Thank you for the opportunity to review your exciting research. David Blodgett.

Reviewer #3 (Remarks to the Author):

The manuscript is an impressive amount of work, with multiple data sets and numerous multi-omics analyses, to characterize the human beta cell response to treatment to IFNalpha, which is expressed in the islets of T1D individuals, and shown to be regulated by genetic and environmental factors associated with T1D.

We thank the Reviewer for her/his positive comments, and for the valuable suggestions for improvement.

General

1. *The manuscript is very dense with a lot of information but there seems to be some missing motivation (see below under Study Design), and overview information (e.g., see below under analysis regarding gene-protein overlap).*

We have now taken action to better clarify the motivation of the study and to provide a more detailed overview information. This new information is provided on Introduction, page 03.

2. It is not clear what the novelty of this contribution. Is it the type of cells? the types of analysis? The authors should motivate how this work is unique.

Type I IFN signaling has many cell-specific components which are mediated by differences in cell surface receptors expression, activation of downstream kinases and the pattern of transcription factors activation (see ^{26,27} and references herein). Thus, and considering the relevance of this cytokine in the pathogenesis of T1D, it is crucial to characterize its effects on human beta cells. Furthermore, the proposed integrated multi-omics analysis has never been done before for IFNα-treated cells. This is now better explained at the Introduction.

3. Why are all data sets not collected at all 3 time points (e.g., proteomics missing at 2h, ATACseq missing at 8h).

Chromatin structure dynamics are critical to changes in gene expression. As such, we hypothesized that changes in open chromatin would precede changes in both RNA expression and protein abundance. We thus reasoned that an early (2h) and a late (24h) data points would be sufficient to obtain an initial understanding of the open chromatin dynamics after $\text{INF}\alpha$ exposure. In fact, our data shows that most of the chromatin accessibility gains occur at 2 hours and are stabilized by 24 hours. Importantly, OCRs gained at 2h are more tightly associated with up-regulation of the nearest genes at 8h rather than gene activation at 2h, supporting the hypothesis that chromatin remodeling precedes gene activation.

4. Why are the 2h, 8h and 24h time points the most relevant, how were they selected?

The time points were selected based on our previous studies where we have performed time series experiments to determine the activation of some transcription factors downstream of the type I interferon receptor and the upregulation of their targets mRNAs and proteins⁴ (see **Figure R10**). These targeted studies allowed us to identify 2h, 8h and 24h as representative time point for early, intermediary and late responses to $\text{INF}\alpha$. This information is now included in Supplementary Methods, page 1.

[REDACTED]

Analysis

5. A lot of the analyses is based on the RNA-Seq data. The authors use RPKM as normalized sequence count values. However, RPKM has been recognized to have major weaknesses, even by the authors of RPKM, it is now more common to use methods such as TPM (or TMM). Do your conclusions change if one of these alternatives are used instead?

For the differential expression analysis of genes and transcripts we do not expect an impact of choosing RPKM as the method to represent gene expression, since we have used for these comparisons the raw read counts obtained from Flux Capacitor. These read counts were then normalized by using the Trimmed Mean of M-values (TMM) method in EdgeR (version 3.26.8)²⁸, which calculates a set of normalization factors, one for each sample, and try to eliminate composition biases between libraries. The data and statistics presented in **Supplementary Tables S3, 5, 6 and 7** were generated using this normalization approach. We have now expanded the differential expression analysis description in the Supplementary Methods adding this information (see page 02). Furthermore, in order to decrease the possible impact of individual sample variability we have used the RPKM average of the different conditions to consider a gene as expressed in our datasets (see page 31). In the WGCNA analysis we have used \log_2 -transformed RPKM values $\log_2(x+1)$. Regarding this point, the authors of WGCNA have previously stated:

*“Whether one uses RPKM, FPKM, or simply normalized counts doesn't make a whole lot of difference for WGCNA analysis as long as all samples were processed **the same way**. These normalization methods make a big difference if one wants to compare expression of gene A to expression of gene B; but WGCNA calculates correlations for which gene-wise scaling factors make no difference. (Sample-wise scaling factors of course do, so samples do need to be normalized.)”*, from:

<https://horvath.genetics.ucla.edu/html/CoexpressionNetwork/Rpackages/WGCNA/faq.html>, last accessed on 08/02/2020.

Finally, RPKM values were selected for representing the gene expression to allow comparison between the present results and our previously published RNA-seq of human islets and EndoC- β H1 cells^{1,29,30}, which were performed using similar techniques and coverage.

6. What is the overlap between the genes/protein products that are measured using RNA-Seq and proteomics, and overlap with the transcripts/proteins that are significantly changing?

We present below in **Figure R11** the requested information on the overlap. As previously described, we observed that these methods detected approximately 10,000 mRNAs/protein in common. As described in **Supplementary Fig. S4**, the overlap between differentially expressed mRNAs and proteins is higher among the upregulated than the downregulated ones.

Figure R11. Overlap between RNA-seq and proteomics datasets. **A.** The overlap between all protein-coding mRNAs identified in the present RNA-seq of EndoC- β H1 cells (mean RPKM > 0.5 in at least one condition) and the proteomics dataset. **B.** The overlap between upregulated protein-coding mRNAs (FC > 1.5 and FDR < 0.05) and upregulated proteins of EndoC- β H1 cells exposed to IFN α (FC > 1 and p-value < 0.05). **C.** The overlap between downregulated protein-coding mRNAs (FC < 0.5 and FDR < 0.05) and downregulated proteins of EndoC- β H1 cells exposed to IFN α (FC < 1 and p-value < 0.05).

7. It is not clear if WGCNA was run with all time points & treated/untreated samples. If so, the sample size appears to be 20 for the RNA-Seq (n=5 X 2 time points X 2 treatments = 20) and 16 for the protein (n=4 X 2 time points X 2 treatments). This is on the low end to run a method like WGCNA, and there are concerns about the stability of the results. Did the authors do any sensitivity analysis?

We indeed used all the samples (treated and untreated) from 8 and 24h to generate the WGCNA modules, since for these conditions we have both RNA-seq and proteomics data available to compare. We have now clarified this point on Results, page 13 of the manuscript.

We have evaluated the modules generated during WGCNA analysis by: 1) verifying the quality (or robustness) of the modules from each dataset; 2) checking if the modules are reproduced in independent samples. First, to analyse module quality we have used a set of statistics (density and separability metrics) from the *modulePreservation* function of the R package WGCNA³¹. To measure this, we have re-sampled the dataset 1000-times to create reference and test sets from the original data and evaluate module preservation, represented as the Zsummary for each module, across the resulting networks. Zsummary > 2 indicates moderate preservation and Z > 10 high quality/preservation for each module³¹. These results showed that the modules detected in RNA-seq and proteomics of EndoC- β H1 cells are all well-defined / robust (Zsummary > 10) (new data added to the manuscript; see new **Supplementary Fig. S7D**). Next, we used the same R function to evaluate whether the WGCNA modules from RNA-seq of EndoC- β H1 cells could be found in RNA-seq of human islets. For this comparison we used metrics based on module density and intramodular connectivity to give a composite statistic Zsummary. Interestingly, we observed that 7 out of 8 modules from RNA-seq of EndoC- β H1 cells

exposed to IFN α presented moderate to strong evidence of preservation in RNA-seq of human islets (**Supplementary Fig. S8E**), supporting the reliability of our data.

8. Also for WGCNA, if multiple time points by treatments are used to define co-expression, but what if co-expression is broken due to treatment? Then the resulting WGCNA modules would only be capturing when co-expression is constant across time points and treatments.

Only the time-points 8 and 24h were evaluated in the WGCNA analysis. We have observed that most of the IFN α -induced changes are present at 8h and also observed at 24h (see **Figure 2A**). Considering that IFN α treatment is the source of changes in gene co-expression patterns in our model, we thus expect to capture the majority of these effects in the evaluated datasets.

References:

- 1 Ramos-Rodriguez, M. *et al.* The impact of proinflammatory cytokines on the beta-cell regulatory landscape provides insights into the genetics of type 1 diabetes. *Nat Genet* **51**, 1588-1595, doi:10.1038/s41588-019-0524-6 (2019).
- 2 Eizirik, D. L., Colli, M. L. & Ortis, F. The role of inflammation in insulinitis and beta-cell loss in type 1 diabetes. *Nat Rev Endocrinol* **5**, 219-226, doi:10.1038/nrendo.2009.21 (2009).
- 3 Op de Beeck, A. & Eizirik, D. L. Viral infections in type 1 diabetes mellitus--why the beta cells? *Nat Rev Endocrinol* **12**, 263-273, doi:10.1038/nrendo.2016.30 (2016).
- 4 Marroqui, L. *et al.* Interferon-alpha mediates human beta cell HLA class I overexpression, endoplasmic reticulum stress and apoptosis, three hallmarks of early human type 1 diabetes. *Diabetologia* **60**, 656-667, doi:10.1007/s00125-016-4201-3 (2017).
- 5 Coomans de Brachene, A. *et al.* IFN-alpha induces a preferential long-lasting expression of MHC class I in human pancreatic beta cells. *Diabetologia* **61**, 636-640, doi:10.1007/s00125-017-4536-4 (2018).
- 6 Saunders, D. C. *et al.* Ectonucleoside Triphosphate Diphosphohydrolase-3 Antibody Targets Adult Human Pancreatic beta Cells for In Vitro and In Vivo Analysis. *Cell Metab* **29**, 745-754 e744, doi:10.1016/j.cmet.2018.10.007 (2019).
- 7 Tsonkova, V. G. *et al.* The EndoC-betaH1 cell line is a valid model of human beta cells and applicable for screenings to identify novel drug target candidates. *Mol Metab* **8**, 144-157, doi:10.1016/j.molmet.2017.12.007 (2018).
- 8 Hastoy, B. *et al.* Electrophysiological properties of human beta-cell lines EndoC-betaH1 and -betaH2 conform with human beta-cells. *Sci Rep* **8**, 16994, doi:10.1038/s41598-018-34743-7 (2018).
- 9 Scharfmann, R., Staels, W. & Albagli, O. The supply chain of human pancreatic beta cell lines. *J Clin Invest* **129**, 3511-3520, doi:10.1172/JCI129484 (2019).
- 10 Brozzi, F. *et al.* Cytokines induce endoplasmic reticulum stress in human, rat and mouse beta cells via different mechanisms. *Diabetologia* **58**, 2307-2316, doi:10.1007/s00125-015-3669-6 (2015).
- 11 Marroqui, L. *et al.* TYK2, a Candidate Gene for Type 1 Diabetes, Modulates Apoptosis and the Innate Immune Response in Human Pancreatic beta-Cells. *Diabetes* **64**, 3808-3817, doi:10.2337/db15-0362 (2015).
- 12 Colli, M. L. *et al.* PDL1 is expressed in the islets of people with type 1 diabetes and is up-regulated by interferons-alpha and-gamma via IRF1 induction. *EBioMedicine* **36**, 367-375, doi:10.1016/j.ebiom.2018.09.040 (2018).
- 13 Lundberg, M., Krogvold, L., Kuric, E., Dahl-Jorgensen, K. & Skog, O. Expression of Interferon-Stimulated Genes in Insulitic Pancreatic Islets of Patients Recently Diagnosed With Type 1 Diabetes. *Diabetes* **65**, 3104-3110, doi:10.2337/db16-0616 (2016).
- 14 Juan-Mateu, J. *et al.* SRp55 Regulates a Splicing Network That Controls Human Pancreatic beta-Cell Function and Survival. *Diabetes* **67**, 423-436, doi:10.2337/db17-0736 (2018).
- 15 Mukherjee, N. *et al.* Integrative regulatory mapping indicates that the RNA-binding protein HuR couples pre-mRNA processing and mRNA stability. *Mol Cell* **43**, 327-339, doi:10.1016/j.molcel.2011.06.007 (2011).
- 16 Bruun, G. H. *et al.* Global identification of hnRNP A1 binding sites for SSO-based splicing modulation. *BMC Biol* **14**, 54, doi:10.1186/s12915-016-0279-9 (2016).
- 17 Ackermann, A. M., Wang, Z., Schug, J., Naji, A. & Kaestner, K. H. Integration of ATAC-seq and RNA-seq identifies human alpha cell and beta cell signature genes. *Mol Metab* **5**, 233-244, doi:10.1016/j.molmet.2016.01.002 (2016).

- 18 Arda, H. E. *et al.* A Chromatin Basis for Cell Lineage and Disease Risk in the Human Pancreas. *Cell Syst* **7**, 310-322 e314, doi:10.1016/j.cels.2018.07.007 (2018).
- 19 Thurner, M. *et al.* Integration of human pancreatic islet genomic data refines regulatory mechanisms at Type 2 Diabetes susceptibility loci. *Elife* **7**, doi:10.7554/eLife.31977 (2018).
- 20 Zhang, Y. *et al.* Model-based analysis of ChIP-Seq (MACS). *Genome Biol* **9**, R137, doi:10.1186/gb-2008-9-9-r137 (2008).
- 21 Xia, C. Q. *et al.* Increased IFN-alpha-producing plasmacytoid dendritic cells (pDCs) in human Th1-mediated type 1 diabetes: pDCs augment Th1 responses through IFN-alpha production. *J Immunol* **193**, 1024-1034, doi:10.4049/jimmunol.1303230 (2014).
- 22 Dotta, F. *et al.* Coxsackie B4 virus infection of beta cells and natural killer cell insulinitis in recent-onset type 1 diabetic patients. *Proc Natl Acad Sci U S A* **104**, 5115-5120, doi:10.1073/pnas.0700442104 (2007).
- 23 Winter, P. M. *et al.* Proinflammatory cytokines and chemokines in humans with Japanese encephalitis. *J Infect Dis* **190**, 1618-1626, doi:10.1086/423328 (2004).
- 24 Antonelli, A. *et al.* Serum levels of proinflammatory cytokines interleukin-1beta, interleukin-6, and tumor necrosis factor alpha in mixed cryoglobulinemia. *Arthritis Rheum* **60**, 3841-3847, doi:10.1002/art.25003 (2009).
- 25 Frankish, A. *et al.* GENCODE reference annotation for the human and mouse genomes. *Nucleic Acids Res* **47**, D766-D773, doi:10.1093/nar/gky955 (2019).
- 26 Schreiber, G. The molecular basis for differential type I interferon signaling. *J Biol Chem* **292**, 7285-7294, doi:10.1074/jbc.R116.774562 (2017).
- 27 Urin, V., Shemesh, M. & Schreiber, G. CRISPR/Cas9-based Knockout Strategy Elucidates Components Essential for Type 1 Interferon Signaling in Human HeLa Cells. *J Mol Biol* **431**, 3324-3338, doi:10.1016/j.jmb.2019.06.007 (2019).
- 28 Robinson, M. D., McCarthy, D. J. & Smyth, G. K. edgeR: a Bioconductor package for differential expression analysis of digital gene expression data. *Bioinformatics* **26**, 139-140, doi:10.1093/bioinformatics/btp616 (2010).
- 29 Eizirik, D. L. *et al.* The human pancreatic islet transcriptome: expression of candidate genes for type 1 diabetes and the impact of pro-inflammatory cytokines. *PLoS Genet* **8**, e1002552, doi:10.1371/journal.pgen.1002552 (2012).
- 30 Gonzalez-Duque, S. *et al.* Conventional and Neo-antigenic Peptides Presented by beta Cells Are Targeted by Circulating Naive CD8+ T Cells in Type 1 Diabetic and Healthy Donors. *Cell Metab* **28**, 946-960 e946, doi:10.1016/j.cmet.2018.07.007 (2018).
- 31 Langfelder, P., Luo, R., Oldham, M. C. & Horvath, S. Is my network module preserved and reproducible? *PLoS Comput Biol* **7**, e1001057, doi:10.1371/journal.pcbi.1001057 (2011).

REVIEWERS' COMMENTS:

Reviewer #1 (Remarks to the Author):

The authors have responded to a large proportion of my comments. However, based on addition of new data some additional concerns have emerged. Please see below.

1. Have the authors checked if the "so called" candidate genes for T1D (included in Sup Fig S2), identified from GWAS, are linked to the T1D associated SNPs through eQTLs in islets? Otherwise, how can they be sure it is the "correct" gene? This is unclear.
2. The new data in figure S6 is nice. Did the authors check viability of the FACS sorted human beta cells prior to the experiments? Also the house keeping gene, beta-actin, may be regulated. Did you check another house keeping gene in the primary human beta cells?
3. I miss p and q values in sup tables, for example the one showing gained OCRs.

Reviewer #2 (Remarks to the Author):

Summary: the authors offered detailed responses to reviewer comments and made appropriate changes to make the manuscript more organized, clearer, and descriptive, which better presents their data in context of the biology. Their cell line experimentation and human islet validation of the multiple-omics approaches are interesting on two levels. First, impact of cytokines on beta cell expression patterns are informative for understanding how the islet (and beta cells) responds to autoimmune attack. Second, in more general way, their study integrating multiple data sources to provide a systematic overview of what happens in the epigenome, transcriptome, and proteome. They show how all of these big data sources can be analyzed together in order to link genotype and phenotype and highlight how genetic changes have an impact at the functional protein level. The comments below can be characterized as suggestions for additional clarity, as the experiments and data presentation comprise a publication-worthy manuscript.

Figure 1: In G, may be helpful to say "8h/24h Upregulated" over the red plots (left) and "8h/24h Downregulated" over the blue plots (right). Supplementary Figure 2 – it would be helpful to label D/E as EndoC, since F/G are labeled as HI. The authors commented on the poor correlation between RNA and Protein in the downregulated samples (lines 199-201 on page 8). Do the authors have any comment re: why there are so many more upregulated regions (1B)/genes (1G) than downregulated regions? (Observed in lines 165-169).

Figure 6: It would be helpful here to indicate which plots are EndoC and which are human islets cells. The figure legend is quite dense, and since they don't progress in an order, adding the HI to appropriate plots would assist the reader.

Reviewer #3 (Remarks to the Author):

My comments were adequately addressed.

REVIEWERS' COMMENTS:

Reviewer #1 (Remarks to the Author):

The authors have responded to a large proportion of my comments. However, based on addition of new data some additional concerns have emerged. Please see below.

1. Have the authors checked if the "so called" candidate genes for T1D (included in Sup Fig S2), identified from GWAS, are linked to the T1D associated SNPs through eQTLs in islets? Otherwise, how can they be sure it is the "correct" gene? This is unclear.

Answer: The candidate genes for T1D were selected from the GWAS based on the following added criteria:

- 1) T1D as the Disease/Trait evaluated by the study;**
- 2) A p-value for the lead SNP $< 0.5 \times 10^{-8}$;**
- 3) Selecting the reported risk genes linked to the lead SNP by the original study;**
- 4) Filtering only the reported genes expressed in human islets (mean RPKM > 0.5 at basal condition or after exposure to $IL1\beta+IFN\gamma^1$) for functional enrichment analysis**

The definition of the target genes regulated by the risk SNPs is indeed an important point. The eQTLs are dynamic² and the identification of the risk variant modulated genes ideally would require that they are evaluated under similar conditions as the ones prevailing during disease, in this case an inflammatory environment. In agreement with this, we have recently demonstrated that the frequency of risk variants in human islets linked to a regulatory element (RE) is increased several-fold by the exposure to the proinflammatory cytokines ($IL1\beta+IFN\gamma$)³. There are limited publicly available databases of eQTLs in pancreatic islets at basal conditions^{4,5}, but there is no information about eQTL in human islets exposed to inflammatory conditions, which prevents us from using eQTL as the selection criteria for the risk genes.

We have now included the complete criteria for T1D risk genes selection to Supplementary Methods.

2. The new data in figure S6 is nice. Did the authors check viability of the FACS sorted human beta cells prior to the experiments? Also the house keeping gene, beta-actin, may be regulated. Did you check another house keeping gene in the primary human beta cells?

Answer: During the FACS gating procedures we remove debris, doublets and dead cells, and use only the viable cells (60% viable cells under control condition and 62% viable cells following $IFN\alpha$ -treatment; $n=3$). The FACS-sorted cells are immediately used for mRNA isolation, without any additional period in culture. In other words, the cells are exposed to cytokines as whole islets, where the viability under control condition or $IFN\alpha$ treatment is around 90%, then dispersed and FACS-sorted for the mRNA extraction only.

We have now analysed a second housekeeping gene (GAPDH) in FACS-purified human beta cells and observed similar results as compared to when using beta-actin as a housekeeping gene (see Figure R1 below).

A**B**
Figure R1. (for the Reviewers only). Comparison of housekeeping genes. FACS-purified human beta cells (A and B) were exposed or not to IFN α for 24h and mRNA expression of HLA-I (ABC), CXCL10, MDA5, CHOP, HLA-E, and MX1 was evaluated by real-time RT-PCR. The values were normalized by two different housekeeping genes: Beta-actin (**A**) or GAPDH (**B**), and then by the highest value of each experiment considered as 1 (n = 4, **p < 0.01, ***p < 0.001, two-sided paired *t* test).

3. I miss p and q values in sup tables, for example the one showing gained OCRs.

Answer: This information was now added to the Supplementary Data 2.

Reviewer #2 (Remarks to the Author):

Summary: the authors offered detailed responses to reviewer comments and made appropriate changes to make the manuscript more organized, clearer, and descriptive, which better presents their data in context of the biology. Their cell line experimentation and human islet validation of the multiple-omics approaches are interesting on two levels. First, impact of cytokines on beta cell expression patterns are informative for understanding how the islet (and beta cells) responds to autoimmune attack. Second, in more general way, their study integrating multiple data sources to provide a systematic overview of what happens in the epigenome, transcriptome, and proteome. They show how all of these big data sources can be analyzed together in order to link genotype and phenotype and highlight how genetic changes have an impact at the functional protein level. The comments below can be characterized as suggestions for additional clarity, as the experiments and data presentation comprise a publication-worthy manuscript.

Figure 1: In G, may be helpful to say “8h/24h Upregulated” over the red plots (left) and “8h/24h Downregulated” over the blue plots (right). Supplementary Figure 2 – it would be helpful to

label D/E as EndoC, since F/G are labeled as HI. The authors commented on the poor correlation between RNA and Protein in the downregulated samples (lines 199-201 on page 8). Do the authors have any comment re: why there are so many more upregulated regions (1B)/genes (1G) than downregulated regions? (Observed in lines 165-169).

Answer: We are grateful to the Reviewer for the positive comments. As suggested, we have now included the suggested additional information in Figure 1G and Supplementary Figure 2.

In respect to the differences between down- and up-regulated regions/genes, we have evaluated cells exposed to IFN α for a maximum period of 24h. We have identified that the highest number of upregulated transcripts is present at 8h, while the number of down-regulated transcripts increased at 24h (see Figure 5A), which is in line with the fact that first “closed” chromatin regions were observed only at 24h (Figure 1B). This suggests that the transcriptional down-regulation of genes is a later phenomenon, probably associated with the progressive beta cell dysfunction. Another possible mechanism, independently of changes in chromatin accessibility, is that expression of some mRNAs are downregulated by the lack of regulatory factors (transcriptional factors, co-activators, etc) as previously described during oocytes reprogramming⁶. The early chromatin opening and gene up-regulation is probably related to beta cells early “cell-based innate immune” responses, aiming to control a putative viral infection⁷.

Figure 6: It would be helpful here to indicate which plots are EndoC and which are human islets cells. The figure legend is quite dense, and since they don't progress in an order, adding the HI to appropriate plots would assist the reader.

Answer: We agree with the Reviewer and have now divided the original Figure 6 in three new figures (Figures 7, 8 and 9) allowing the reader to better identify the cell type used in each experiment.

Reviewer #3 (Remarks to the Author):

My comments were adequately addressed.

References:

- 1 Eizirik, D. L. *et al.* The human pancreatic islet transcriptome: expression of candidate genes for type 1 diabetes and the impact of pro-inflammatory cytokines. *PLoS Genet* **8**, e1002552, doi:10.1371/journal.pgen.1002552 (2012).
- 2 Strober, B. J. *et al.* Dynamic genetic regulation of gene expression during cellular differentiation. *Science* **364**, 1287-1290, doi:10.1126/science.aaw0040 (2019).
- 3 Ramos-Rodriguez, M. *et al.* The impact of proinflammatory cytokines on the beta-cell regulatory landscape provides insights into the genetics of type 1 diabetes. *Nat Genet* **51**, 1588-1595, doi:10.1038/s41588-019-0524-6 (2019).
- 4 Varshney, A. *et al.* Genetic regulatory signatures underlying islet gene expression and type 2 diabetes. *Proc Natl Acad Sci U S A* **114**, 2301-2306, doi:10.1073/pnas.1621192114 (2017).

- 5 Fadista, J. *et al.* Global genomic and transcriptomic analysis of human pancreatic islets reveals novel genes influencing glucose metabolism. *Proc Natl Acad Sci U S A* **111**, 13924-13929, doi:10.1073/pnas.1402665111 (2014).
- 6 Miyamoto, K. *et al.* Chromatin Accessibility Impacts Transcriptional Reprogramming in Oocytes. *Cell Rep* **24**, 304-311, doi:10.1016/j.celrep.2018.06.030 (2018).
- 7 Op de Beeck, A. & Eizirik, D. L. Viral infections in type 1 diabetes mellitus--why the beta cells? *Nat Rev Endocrinol* **12**, 263-273, doi:10.1038/nrendo.2016.30 (2016).